# Automated Porosity Characterization for Aluminum Die Casting Materials Using X-ray Radiography, Synthetic X-ray Data Augmentation by Simulation, and Machine Learning

**DOI:** 10.3390/s24092933

**Published:** 2024-05-05

**Authors:** Stefan Bosse, Dirk Lehmhus, Sanjeev Kumar

**Affiliations:** 1Department of Mathematics & Computer Science, University of Bremen, 28359 Bremen, Germany; 2Fraunhofer Institute for Manufacturing Technology and Advanced Materials, 28359 Bremen, Germany; dirk.lehmhus@ifam.fraunhofer.de; 3Department of Mechanical Engineering, University of Bremen, 28359 Bremen, Germany

**Keywords:** simulation, machine learning, synthetic data augmentation, X-ray, porosity analysis

## Abstract

Detection and characterization of hidden defects, impurities, and damages in homogeneous materials like aluminum die casting materials, as well as composite materials like Fiber–Metal Laminates (FML), is still a challenge. This work discusses methods and challenges in data-driven modeling of automated damage and defect detectors using measured X-ray single- and multi-projection images. Three main issues are identified: Data and feature variance, data feature labeling (for supervised machine learning), and the missing ground truth. It will be shown that simulation of synthetic measuring data can deliver a ground truth dataset and accurate labeling for data-driven modeling, but it cannot be used directly to predict defects in manufacturing processes. Noise has a significant impact on the feature detection and will be discussed. Data-driven feature detectors are implemented with semantic pixel Convolutional Neural Networks. Experimental data are measured with different devices: A low-quality and low-cost (Low-Q) X-ray radiography, a typical industrial mid-quality X-ray radiography and Computed Tomography (CT) system, and a state-of-the-art high-quality μ-CT device. The goals of this work are the training of robust and generalized data-driven ML feature detectors with synthetic data only and the transition from CT to single-projection radiography imaging and analysis. Although, as the title implies, the primary task is pore characterization in aluminum high-pressure die-cast materials, but the methods and results are not limited to this use case.

## 1. Introduction

High-Pressure Die Casting (HPDC) is an established manufacturing process for large-scale series production affording the highest productivity. The main materials for which it is employed are aluminum alloys, followed by magnesium and zinc. Until recently, typical HPDC machine sizes did not exceed 4.5 k tons. Nowadays, since the automotive OEM Tesla initiated Gigacasting using 6 k tons equipment supplied by Italian manufacturer IDRA, 12 k tons machines have been realized to produce battery housings, and 18–20 k tons systems are under development for single-shot production of the combined front and rear underbody plus battery housing of passenger vehicles. This, on the other hand, raises questions about defect-related variability of properties: In a typical HPDC process, mold filling happens at extremely high melt velocities within tens of milliseconds and is thus extremely turbulent. Additionally, feeding is limited, as it can only be realized via the plunger and thus becomes ineffective as soon as the in-gates are solidified. As a consequence, porosity caused by shrinkage of an isolated melt volume or by entrapped gas is likely to occur, weakening the material properties. For this reason, for safety-relevant components, the industry has preferred low-pressure die casting (LPDC) in the past [1,2]. However, with the advent of Gigacasting, the contribution of HPDC parts to crashworthiness gains further importance, while producing castings of the sizes mentioned above is an extremely challenging task anyway, and may afford compromises in terms of lightweighting for processability alone, notwithstanding the potential need to raise safety factors if defects can neither be avoid nor detected with sufficient accuracy [3]. Despite this, development of Gigacasting technology is now being extended to magnesium alloys, too, in search of additional automotive lightweighting potentials [4]. This situation motivates the current study, which is aimed at improving detection and classification of casting defects via non-destructive approaches.

In general, such defects can be identified in two ways: On the one hand, relevant aggregative material and defect characteristics can be measured using destructive methods like tensile tests and micrograph slicing but only giving some sample results. On the other hand, Non-destructive Testing (NDT) encompasses a broad range of methods to detect and characterize material properties, defects, impurities, and damages in a wide range of materials and structural composites, e.g., laminates and composite materials. However, NDT based on imaging methods can only deliver geometrical properties. The majority of NDT methods are based on wave propagation and interaction, i.e., from electromagnetic fields, light fields (visible and invisible), sound and vibration fields, and X-ray radiation fields. In this work, we only focus on X-ray radiation-based methods, basically radiography (producing single-projection images) and Computed Tomography (CT), based on multiple radial image projections, finally deriving volume slices by means of reconstruction methods [5]. We thus concentrate on image processing algorithms and feature detection in images (i.e., features representing damage and defects) using pure data-driven models.

Therefore, physics-based simulation should be used to retrieve relevant information about critical process conditions resulting in defects and their characteristics. Because any simulation is based on simplified models, the simulation results should be compared with experimentally measured material properties and defect characterizations.

Feature detection and marking of defects in measuring images can occur on different levels:Region-of-Interest Search (marking, there is something somewhere maybe);Feature Maps (marking, localized features);Damage and defect classification;Damage and defect localization;Global statistical aggregates (e.g., pore density, distributions, etc.).

Either classical numerical and model-based algorithms (e.g., based on edge detection using a Soebel filter or Canny detector) or data-driven models are used for feature marking (Machine Learning models), e.g., based on shape and object classifiers like Convolutional Neural Networks (CNN) or more advanced models like region-proposal networks (ResNet [6], YOLO [7], SAM [8]). Any data-driven model can be considered as a parametrized function *F*(*P*,*X*): *X* → *Y* that maps an input data space *X* commonly derived from measuring data or already computed features on an output data space *Y*, commonly defect or damage features. The parameter set *P* must be approximated using training data.

The primary goal of this work is automated damage, defect, and impurity detection in materials including composites using single-projection X-ray images and data-driven feature marking models (e.g., deploying Convolutional Neural Networks). Detection of hidden damages, defects, and impurities like gas pores is still a challenge using X-ray radiography. X-ray images typically pose a low contrast if the density of defects is close to the host material or if they are very small compared to the resolution of the measuring device. It can be shown that classical image processing algorithms are not suitable for accurate pore characterization in X-ray radiography images. Instead, we are proposing and evaluating data-driven trained predictor Machine Learning (ML) models.

Data-driven models require data with a sufficient statistical variance and distribution of features to be detected. That is the first issue with most engineering data lacking variance and posing a low degree of representation (weak database). Additionally, supervised training of high-dimensional and parametrized models requires accurately labeled (annotated) strong feature examples, commonly not available. This is the second issue and downfall in data-driven modeling. Both aspects can be summarized as the missing ground truth issue. The combination of simulated (synthetic) and real-world data is commonly required. For example, Mery et al. [9] addressed aluminum die casting inspection using complex deep learning models and simulation of simplified ellipsoidal effects. Their approach relied on the superposition technique, overlaying real X-ray images with simulated parts of defects.

The novelties and topics of the present study are (see also Figure 1) a fusion of multiple methods:Physics-based simulation delivering basic information about defects in die casting processes used for the qualitative validation of defect recognition results.Material- and structural model-based simulation and synthetic sensor data (SD) generation;Filling the ground truth gap in real sensor data (RD) by using accurately annotated data provided by CAD modeling;Training of data-driven ML models only using SD, but prediction (inference) on RD;Predictor models with low complexity and divide-and-conquer approaches for feature class marking in images;Point/pixel clustering methods for object reconstruction and shape fitting, finally performing defect characterization and statistical analysis.

**Figure 1 sensors-24-02933-f001:**
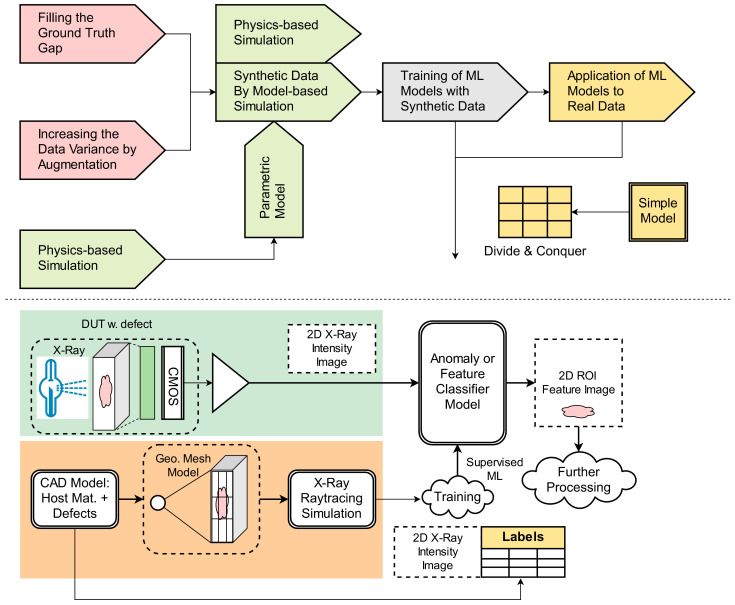
(**Top**) Methodologies used in this work. (**Bottom**) Data and model architecture.

Considering small defects and damages in homogeneous or composite materials, there is no experimental ground truth method to characterize such defects like gas pores with high accuracy and trust. Micrograph slicing is a common technique to characterize pores, but this method is limited to 2D views, and the slicing process alters the material and the defects, e.g., due to polishing or etching processes. High-resolution μ-CT techniques are suitable to characterize small defects, but the mathematical functions and algorithms used to reconstruct 3D volume slices from radial projections introduce artifacts and limitations that require pre- and post-filtering, finally reducing the accuracy and trust of CT images. Besides, high resolution in the typical size range of gas porosity in castings is only available for extremely small volumes. In conclusion, there is no experimental ground truth training dataset for the supervised training of data-driven feature (defect) marking models. We propose a unified simulation flow to generate synthetic data used exclusively for training of predictor models. We will show that these models can be applied to real measurement images. We will also show that a model-based physical simulation of a die-casting process is not sufficient to predict certain aspects of porosity as observed by experiments but can be used to obtain basic knowledge and information for the structural modeling of materials with defects.

In [7], a similar approach of X-ray data augmentation and feature marking was considered, also using the superposition technique for the generation of synthetic image data, i.e., creating a superposition of real images (without defect) with “simulated” defect image fragments. In doing so, ref. [7] uses an established, but very complex, detector model (YOLO) for defect feature marking in X-ray images. In contrast, we use very simple models (semantic pixel classifier with flat CNNs) and a divide-and-conquer approach. Furthermore, the types of defects studied by [7] appear to be geometrically simple, while our own approach can in principle be transferred to any host material and defect geometry. While in the present study, we primarily focus on pore classification—a rather simple geometric task—in the outlook, we will present the deployment of our approach for (impact) damage detection in and characterization of multi-layered Fiber–Metal Laminate (FML) plates. Even though there are more problems than solutions regarding FML impact damage modeling, this use case serves to underpin the power of our simulation approach.

Via this technique, we analyze pores commonly created during the high-pressure die-casting process. There are different classes of pores, summarized, e.g., as gas or shrinkage porosity, showing different geometries and size ranges. We will provide a methodology to generate synthetic measurement data covering the aforementioned classes relying entirely on simulation. These synthetic data are used to train data-driven feature marking models, finally providing a statistical analysis of pores by using point clustering and geometrical shape fitting.

Synthetic data generation is detailed in section Section 3, which introduces the synthetic data generation by simulation, followed by 3D Computed Tomography (CT) analysis providing reference data for simulation, and finally the feature marking models, being the main major contributions of this work. High-Pressure Die Casting (HPDC) and its manufacturing defects is the main use case we will consider in this work. The HPDC process simulation provides some insights into porosity. To understand the challenges and issues with defect detection in this use case, the following Section 2 introduces the basics of HPDC manufacturing.

## 2. High-Pressure Die Casting

The typical part weights realized range from grams up to several 10 kg. The primary customer specifically for aluminum HPDC parts is the automotive industry, with typical components found in powertrains based on internal combustion engines. The current transition from ICE to battery electric vehicles (BEV) has eliminated components like cylinder blocks or gear boxes, forcing suppliers of castings to focus on structural applications. A recent development in this respect is Giga- or Megacasting, which has further extended shot weights to beyond 100 kg: Tesla has initiated this trend by producing the rear underbody of their Model Y as single-piece casting, claiming cost advantages based on the replacement of several sheet metal components, plus associated joining operations. By now, several automotive OEMs such as Volvo, Nio, XPeng, and others are taking up this technology, while equipment manufacturers like Bühler or Haitian are following IDRA and LK Machinery in offering ever-larger HPDC machines [10,11]: Tesla started their respective activities using IDRA equipment offering locking forces of 6000 tons, exceeding the then-maximum of roughly 4500 tons by a third.

### 2.1. Porosity in HPDC

While HPDC excels in productivity, it is distinguished from other casting processes by the fact that mold filling is extremely fast and turbulent—more of a spraying than a flow-based process. Melt quality, turbulence, and solidification processes may all contribute to the occurrence of porosity in an HPDC component. Casting simulation is capable of providing insights into areas specifically prone to such defects but can typically not predict the actual expression of porosity in terms of measures like average or maximum pore size, much less pore size distribution [12,13]. Similarly, the effects of porosity on part properties are not known in all possible detail: When it comes to guaranteeing mechanical properties, automotive OEMs thus tend to rely on specifying maximum allowable pore sizes either on a general level, or for critically loaded regions of the respective component.

Pore sizes and shapes tend to differ depending on the origin of the pores. Gas porosity is typically rounded and small in size, while shrinkage pores are larger and tend to be of irregular shape, as their inner surface reflects the formation of crystallites during solidification. Given the aforementioned geometric features, and provided the respective data are available, a distinction is possible via criteria like surface-to-volume ratio and similar characteristics relating the actual pore volume to that of an enclosing sphere or ellipsoid [14].

In general, gas pores tend to be less critical in terms of property deterioration than, e.g., shrinkage porosity: Their smaller size and spherical shape have been shown to produce less of an adverse effect, e.g., on part strength and elongation than larger shrinkage porosity, the irregular inner surface of which may further kerf effects. Contrary to this general observation, gas porosity may become critical in HPDC components if it is formed at the high levels of intensification pressure (up to and above 1000 bar) which constitute an integral part of the HPDC process, and which serve to ensure feeding as well as pore size reduction. In practice, the conflict between the design engineer focusing on functionality and foundry engineer aiming at castability is often resolved, not by eliminating porosity but by limiting it as far as possible, while at the same time shifting it to low stress regions within the part. The handles to achieve this include processing conditions and mold layout, as well as, the designer permitting, part geometry. The tool with which these measures may be tested is casting simulation, i.e., the numerical prediction of where to expect which types and expressions of defects, accompanied by additional information on the characteristics of the undisturbed matrix. Summing up, based on the nature of the process, economic production, especially of large HPDC components (the Gigacasting approach) without any defects, is next to impossible. Companies engaging in this thus have to make sure they can (a) reliably detect defects and (b) accurately judge their effect on part performance. The current study addresses the first part of these issues, focusing on porosity.

Currently, while X-ray scans are widespread in industry, X-ray-based CT defines the gold standard. Depending on requirements, the frequency of scans ranges from a per-shift to an individual part basis. Furthermore, scans of full parts, usually with reduced resolution, and detailed scans of critical areas can be distinguished. Similarly, different practices exist in terms of the actual identification of defects, as well as their classification as critical or not. Typically, the sorting of parts into acceptable and reject categories is carried out manually with the help of reference images, or pore size measurements relying on human recognition. In many OEM specifications, pore size limits are thus the primary decision criteria, with all the consequences of this method, which does not account for other characteristics of a specific porosity distribution. Alternatives, which rely on specifying maximum levels of porosity in highly loaded regions of the casting, also fail in this respect, neglecting aspects like kerf effects, pore size distribution, or spatial arrangement of pores. Automated Defect Recognition (ADR) could provide just this by deriving further defect characteristics beyond the fundamental: It would even allow storage of metadata on each individual part, for documentation, or for clarification of any in-service failure, should that occur—and thus also for adjustments in evaluation procedures. Furthermore, understanding of the effects of defects could benefit from unambiguous, in-depth characterization of pore size distributions: Applied to samples for mechanical testing, the method could provide a basis for new insights in this respect as well as correlations between process parameters and part properties. Finally, e.g., via stochastic simulation approaches as discussed by Andrieux et al. [15,16], this could help solve the problem of properly dimensioning HPDC components expressed by Blondheim and Monroe: “The general region where macro porosity forms is predictable with simulation, but its actual size and distribution of the voids are random” [13].

### 2.2. Influence of Porosity on Part Properties

Porosity is well known to have a detrimental influence on part performance, lowering yield strength, ultimate tensile strength, and elongation at failure, i.e., ductility. Studies on these subjects are numerous and have, e.g., been published by several authors both for HPDC, as well as for LPDC, and considering various types of loads, i.e., static, dynamic, or cyclic (see e.g., [17,18]). Zhang et al. compared HPDC and gravity die-cast samples of an AlSi7MnMg alloy, concluding that shrinkage porosity is dominant when it comes to property reduction in comparison with entrapped gas. Furthermore, they demonstrate the influence of maximum pore size by contrasting samples with 0.3 mm and 1.3 pores, of which the former reach 9–13.5% elongation at failure, while the latter fail at 6.4% elongation, and claim an inverse proportionality between both parameters. In addition, they introduce a limit value of sphericity at 0.4 in order to distinguish between gas and shrinkage porosity, with the latter associated with sphericity values at or below this limit [19]. Another group of researchers extends the aforementioned studies to magnesium alloys, which are typically less ductile due to their hcp lattice structure vs. the fcc of aluminum alloys, using X-ray CT to non-destructively characterize the samples prior to mechanical testing. Zhang et al., in their work, distinguish between a pore fraction of smaller size (equivalent diameter at or below 90 µm), which dominates in number, but contributes only 50 vol.%, and larger size pores above 425 µm equivalent diameter, associated with 5 vol.% of overall porosity. In order to include, to a limited degree, the spatial distribution of porosity, they divide their samples into several slices, with the (virtual) cutting plane oriented perpendicular to the direction of the load in tensile testing. By this means, they can in principle establish an improved correlation between porosity and mechanical properties, especially Ultimate Tensile Strength (UTS), provided the samples fail in the assumed region. Their experimental data confirm good correlation mainly between critical section porosity and elongation at failure [20]. Yu et al., who also work on magnesium HPDC, go beyond the former study by focusing even more on aspects of defect morphology and microstructure, including, e.g., local variation of the latter in terms of externally solidified crystals. Interestingly, they find that the mechanical properties depend on the defect band width rather than the overall level of porosity. Defect bands are defined as local areas of increased defect, i.e., in this case, porosity, level, which may constitute the weakest link within the sample [21].

### 2.3. Detecting Porosity and Deriving Measures to Counter It

For an envisaged in-line process monitoring and control, the HPDC process offers certain parameters which can be adjusted to counter an observed negative trend in porosity characteristics. These could include, e.g., shifting the thermal state of the die either via adaptation of the spraying process or by adjusting volume flow and/or temperature of the cooling/tempering fluids. A further option could be modifying the shot curve, including variation of the intensification pressure. More sophisticated solutions might build on heat pipes for local cooling or squeezers for local feeding, both of which could also be varied in terms of their operational parameters, or on vacuum-assisted or vacuum processes [22,23,24]. A major prerequisite for effectively employing such approaches is the detection of porosity and thus the main topic of the present study. In their study, Nourian-Avval and Fatemi have shown that major differences may be observed when comparing defect data derived from different methods like X-ray CT and μ-CT, radiography and metallographic sections—while the latter appears to be more accurate, it is hampered by its effort and the fact that it is necessarily a destructive technique, whereas the others leave the part under scrutiny intact. The message to be taken from this is that there is an urgent need for improvement in defect recognition based on non-destructive (NDT) solutions [25]. Automated Defect Recognition (ADR) has thus developed into a widely studied topic in this respect, though it does face obstacles. To overcome these, Ji et al. use a two-stage approach employing a filtered selective search algorithm to identify potential defect sites, while categorizing them by means of an Evenly Distributed Convolutional Neural Network (ED-CNN) [26]. Fuchs et al. discuss such concepts, stating that though Convolutional Neural Networks (CNN) and other deep learning methods have successfully been employed to identify defects, they suffer from the drawback of requiring vast amounts of tediously labeled training data. Their suggestion is thus to produce synthetic training data via simulating the tomography process—their claim is that by following this approach, they can provide ground truth data for training on the level of individual voxels. In addition, they compare high- and low-quality CT data, using the latter as an additional basis to verify detection results gained on low-quality data [27]. Mery also proposes the use of synthetic training data to overcome the training dataset issue, and the efforts of labeling, with respect to the analysis of conventional X-ray or radiography data. In two related studies, he compares several deep object identification methods using a hybrid approach which includes generating training data via superimposing real, but defect-free, X-ray with simulated defect data [9,28]. Finally, Hena et al. use synthetic training data generated by distribution modeling of X-ray intensity as training data input for ADR in an approach similar to the one suggested in this study. The difference, however, compared to our own approach, lies [7].

### 2.4. Sample Production

Samples for evaluating advanced ADR techniques were cast from aluminum alloy AlSi9Cu3(Fe) or 226D (see Table 1 below for the composition according to the supplier’s—MMG Aluminum AG, Mayen, Germany—specification) at a temperature of 680 °C using a Frech DAK 250-34 (Schorndorf, Germany) high-pressure die-casting machine with a locking force of 290 tons.

### 2.5. Summary of HPDC Simulation

From the simulation results shown in Section 8.1, the following conclusions can be drawn with respect to the usability and qualitative assessment, as shown in Table 2.

There is no quantitative measurement of porosity that can be derived directly from this simulation, but there are relevant qualitative information and features for the structural CAD-based model design. The qualitative analysis results from the die casting process simulation are mainly used to create coarse pore size distribution splits and spatial weight distributions, discussed in Section 8.1.

## 3. Synthetic Data Generation by Simulation

In this work, data-driven predictor models are applied to X-ray images. These models are trained and supervised by examples. The goal of the predictor model is to mark defects or damages, finally creating feature map images suitable for further defect and damage analysis. The main issue with supervised training is the ground truth of the training examples, which cannot be satisfied in all use cases considered in this work. To overcome the limitations of training data from real measurements with manually annotated labels, we propose to generate synthetic training data with full ground truth by using a unified simulation flow.

The simulation flow consists of the following parts, as summarized in Figure 2:Acquisition and quantitative analysis of a base reference set of defects or damages (here, pore characterization using 3D μ-CT data) and some qualitative results from Section 2.3 (process simulation);Creation of a CAD model consisting of a host material (here, homogeneous aluminum) and defects using Constructive Solid Geometry (CSG) modeling combined with Monte Carlo simulation;Transformation of the CSG-CAD model to a multi-material convex hull mesh-grid model (STL) using the OpenSCAD tool [29];Simulation of X-ray radiography or CT projection images using a GPU-driven ray-tracing simulation (based on the Beer–Lambert law and the gVirtualXray library [30]);Overlaying of noise (additive Gaussian electronics noise, multiplicative Gaussian noise, and Poisson- or binomial-distributed detection process noise).

**Figure 2 sensors-24-02933-f002:**
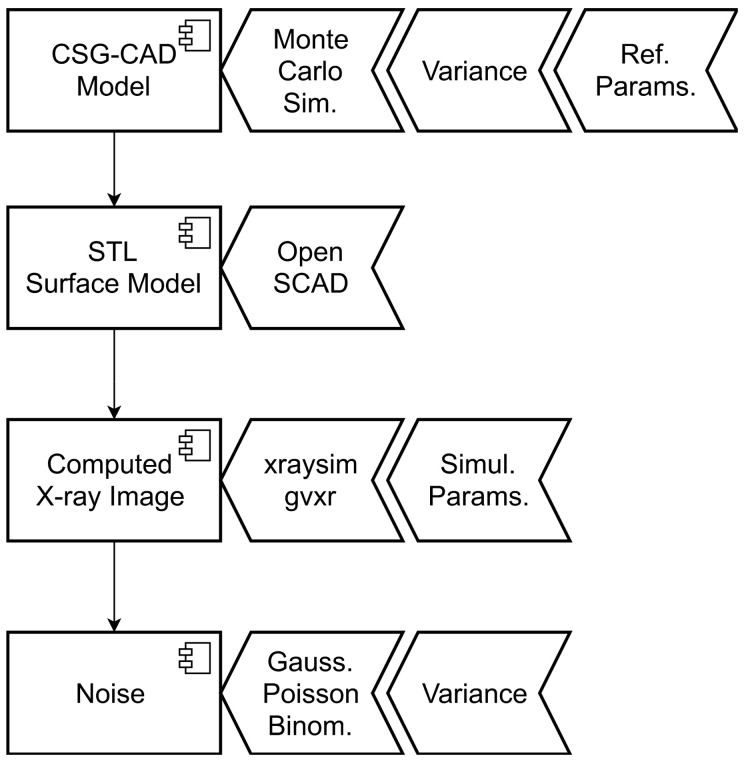
Simulation flow for the generation of synthetic data.

The qualitative analysis results from the HPDC process simulation are mainly used to create coarse pore size distribution splits and geometrical spatial weight distributions. There are basically two classes of pores as derived from HPDC simulation and CT analysis: Small gas pores (diameter below 100 μm) and larger shrinkage pores (diameter above 100 μm).

Monte Carlo simulation was used with a Gaussian random process to introduce variance of geometric and location parameters of defects applied to the measured parameter reference set (from CT analysis and partially from HPDC simulation). The variance parameter σ of the random process was derived from observations and theoretical assumptions.

### 3.1. CSG-CAD Modeling

A parametric mechanical CAD model is required to compute synthetic X-ray images. The CAD model consists basically of two parts:The base-line structure and host material without defects and damages;Defects or damages.

To enable parametric modeling, the CAD model is computed by using Constructive Solid Geometry (CSG) modeling. CSG provides Boolean union, difference, and intersection operations as well as rotation, scaling, and translation operations to construct arbitrarily structured and shaped materials. The complexity of the model and the generation process depends on the host material and the class of defects. In this work, pores are approximated by ellipsoid geometries. A CSG model of an ellipsoidal pore is shown in Algorithm 1. A sphere is scaled to create an ellipsoid. Each pore is modeled independently using Monte Carlo simulation (with statistical normal and uniform distributions) and a reference set of pores computed by a 3D μ-CT analysis from real aluminum die-cast plates.
**Algorithm 1.** Parametrized ellipsoid pore model using CSG operations.**define** pore (xc,…)):**translate**([xc,yc,zc])**rotate** ([xa,ya,za])**scale**([xr,yr,zr])**sphere**(r=0.5,$fn=nsegments);

A plate with pores is created by a subtractive (material difference) operation, as shown in Algorithm 2. The only constraints that must be satisfied are given by position and the surface boundary (pores may not exceed the surface creating holes). Overlapping of pores is allowed, creating composed pore shapes, which can be expected in real physical processes, too. The host material is homogeneous and created by a simple *cube* operation.
**Algorithm 2.** Parametrized plate model with pores using CSG operations.**define** plate (xp,yp,zp,…):**rotate** ([90,90,90])**difference** () {**rotate** ([90,0,0])**cube**([xp,yp,zp],center=true);**union** () {pore(…)pore(…)…};}

A triangular surface mesh-grid model is finally computed from the CSG model and exported into STL format. OpenSCAD [29,31] is based on the OpenCSG and Computational Geometry Algorithm (CGAL) libraries. Special algorithms that can simulate the presentation of Computational Solid Geometry operations (union, intersection, and subtraction) on a two-dimensional screen provide the foundation of OpenCSG. SCS and Goldfeather are its two primary algorithms. CGAL includes a vast array of geometric approaches and algorithms that can be used to represent objects. It facilitates the output of 3D formats such as STL by calculating the actual point sets of 3D objects. Details can be found in [31].

The ordinary Polyhedra and the Nef Polyhedra are the two primary CGAL features that OpenSCAD takes advantage of. It also makes use of a number of additional features, such as triangulation. However, the ordinary Polyhedra and the Nef are the primary data structures.

Non-cubic or rectangular shapes must be approximated by a segmented mesh grid (facets), e.g., for spheres and cylinders. The number of facets (segments) determines the accuracy of computed surfaces. If this surface model is used for X-ray simulation, artifacts and patterns can occur in the computed X-ray images, but practical experience showed no significant impact as long as the surface discretization is below the geometrical resolution limit of the X-ray imaging set-up.

Even complex materials, e.g., Fiber–Metal Laminates, can be modeled accurately, e.g., by creating single fibers by cylinders. A test was performed with an example of a synthetic FML plate with 10,000 glass fibers (embedded in resin) and simplified impact damage deformation, which is shown in Figure 3. The physical size of the test plate was 50 × 50 mm. The impact damage defect can be parametrized with respect to the diameter (about 5–20 mm in the *x*–*y* plane) and the depth (about 0.5–2 mm in the *z*-axis). Using the simulation approach for X-ray images based on the CSG model as discussed in the next section, it can be shown that the micro-scale structure of the fiber matrix is visible in the X-ray images with typical Moiré patterns. The computational time for such a complex triangular mesh-grid transformation performed by OpenSCAD is about 10 min (Dual core Intel Core i5-7300 U—3 GHz max. clock frequency, 8 GB DRAM, embedded Intel HD620 GPU) with 10,000 fibers and depends on the triangular approximation settings (number of segments to approximate circles and ellipses). The X-ray simulation for one image (1000 × 1000 pixels) itself requires less than one second (CPU) or 50 ms (NVIDIA GPU, P4000).

However, from a practical perspective, it is nearly impossible to model fiber materials in a realistic way. Neither the geometrical fiber characteristics nor a statistical model of the placement distribution is known. An escape from this trap is the fusion of real and synthetic data using X-ray images from typical fiber materials (without defects and damages), finally performing a superposition with synthetic images only modeling damages.

To summarize:Homogeneous host materials can be directly and easily modeled, even with complex shapes using poly-line extrusion.Composite materials (e.g., with fibers) are difficult to model and should be replaced with a homogeneous approximation.There are subtractive and additive defects. Additive defects are added on manufacturing, e.g., gas pores, subtractive defects are damages like cracks or delaminations, requiring a constant host material mass and volume constraint (no mass is added or removed).Acquiring reference data for additive defects, especially pores, can be simply carried out by CT analysis and ROI marking. Delaminations and cracks require a higher effort and semi-automated polygon tracking.

### 3.2. X-ray Image Simulation

The X-ray simulation is used to create synthetic X-ray projection images from CAD models. The input is a polygon mesh grid (STL, Stereo lithography file format) model. An STL file describes raw, unstructured triangulated surface. A decomposition of multi-material structures in single-density parts (finally merged in the simulator) is required for composite materials since the STL file format does not support multi-material sections.

X-ray image computation commonly bases on the Beer–Lambert absorption law:(1)$$I\left(x,y\right)=\sum_{i}R\left(E_{i}\right)D\left(E_{I}\right)e^{-\sum_{j}\mu_{j}\left(E_{i}\right)d_{j}\left(x,y\right)}$$

The integrated energy that pixel (*x*,*y*) receives in eV is denoted by *I*(*x*,*y*). The beam spectrum is discretized in one or many energy channels in the poly-chromatic scenario. The energy of the *i*-th energy channel, expressed in eV, is denoted by *E_i_*. The number of photons that the source emits at that energy, *E_i_*, is expressed as *D*(*E_i_*). By substituting a lower value for the incident energy *E*_i_, the detector response *R*(*E_i_*) resembles the operation of a scintillator. Specifically, *R*(*E_i_*) < *E_i_*. The linear attenuation coefficient μ_j_(*E_i_*) is the one of the *j*-th material at energy *E_i_*, with respect to the path length, *d*_j_(*x*,*y*), as shown in Figure 4.

Deterministic simulations based on the Beer–Lambert law generate noise-free images [32]. Depending on the application, they can provide a good compromise between speed and accuracy and can be implemented on GPUs for a further increase in speed. But data-driven algorithms are known to be noise-sensitive, sometimes with unpredictable results. Noise must be added as a post-process, discussed in the next Section.

The gVirtualXray simulation library [30,33] assumes attenuation of X-ray radiation along a straight path. Scattering, energy conversion, and reflection are neglected. On the one hand, this simplification is valid since the major contribution to X-ray radiography images is attenuation. On the other hand, metal materials pose elastic and inelastic scattering, and X-ray scattering alters images even in medical applications. Radial multi-projection measurements with following CT reconstruction are more sensitive to scattering than single-projection radiography, which is only used in this work to obtain reference data for synthetic data generation. Feature analysis is performed by only using radiography images.

### 3.3. Noise

There are multiple sources of noise in X-ray images (according to [34]):X-ray (photon) generation with a statistical Poisson distribution;X-ray attenuation (X-ray Matter Interaction) with a statistical binomial distribution;X-ray detection with a statistical binomial distribution;Additive electronic noise with a statistical Gaussian distribution.

There is additive, multiplicative, and “sampling” noise (e.g., Poisson). Poisson noise is not additive and cannot simply be superimposed to the original data like Gaussian noise, since it is signal-dependent. The Poisson distribution can only be defined for positive integers. To add Poisson or binomial noise to noise-free images requires absolute photon counting during X-ray generation and material interaction, which are not accessible in the final computed X-ray image. To overcome this issue, a normalization factor for the image intensity–photon count relation is used, simulating photon counting in the final image. This is a significant simplification, and noise sources 1 to 3 are accumulated with one statistical distribution.

For instance, the simulated X-ray images of the synthetic plates gave an intermediate output range [50, 80] with a given measuring set-up. The images were scaled to a 16-bit range [0, 65,535] using an intermediate scaling range of [50, 80], i.e., I = 50 → 0, I = 80 → 65,535. The photon conversion factor is therefore γ = (80–50)/65,535 = 4.6; 10^−4^ with an offset correction of 50. The values in the range [0, 80] multiplied with a noise strength factor ω are finally applied to a Poisson-distributed random generator to simulate X-ray noise. Values of *ω* > 1 decrease the noise contribution (higher absolute values), values ω < 1 increase the intensity variance:
(2)$$\begin{array}{l} \omega =\{\begin{array}{cc} <1 & \mathrm{higher}\;\mathrm{noise}\\ >1 & \mathrm{lower}\;\mathrm{noise} \end{array}\\ I_{\mathrm{noisy}}=\frac{\mathrm{Poisson}\left(\lambda =I\cdot \omega \right)}{\omega } \end{array}$$

Binomial random processes require another approach using a sampling probability *p* in the range [0, 1]:(3)$$\begin{array}{l} p=\left[0,1\right]\\ I_{\mathrm{noisy}}=\frac{\mathrm{Binomial}\left(size=I,prob=p\right)}{p} \end{array}$$

Here, the Poisson and binomial functions are vector functions with iterative application to input vector elements *I*. The impact of Poisson and binomial sampling noise on a linear intensity distribution is illustrated in Figure 5 for a linear intensity range [0, 80]. Both statistical distributions show different modification of the respective signal. The Signal-to-Noise ratio (SNR) increases with higher signal values since the number of events, e.g., generated photons, reduces the uncertainty.

Finally, Gaussian noise is added additively (or multiplicatively), simulating basically electronics noise, with some examples shown in Figure 6, showing a continuously increasing Signal-to-Noise ratio:(4)$$\begin{array}{l} I_{\mathrm{noisy+}}=I+\mathrm{Gaussian}\left(1,-\sigma ,\sigma \right)\\ I_{\mathrm{noisy*}}=I\cdot \left(1+\mathrm{Gaussian}\left(1,-\sigma ,\sigma \right)\right) \end{array}$$

Additive noise is independent from the signal, but resulting in an increasing Signal-to-Noise ratio with respect to the signal, in contrast to multiplicative noise with a constant SNR. Additive noise is an extra signal source (similar to artifacts), whereas multiplicative noise is a variance of the respective signal.

As a replacement for Poisson and binomial noise relying on unknown parameters and scales, only Gaussian noise can be added to synthetic data. We used a combination of additive and multiplicative Gaussian noise to enhance the synthetic X-ray image data, which were only used to train the classifier models. In our experiments, we used σ = 0.5 for multiplicative and σ = 200 for additive Gaussian noise augmentation, assuming a digital intensity value range of the synthetic X-ray images of [0, 65,535]. In Section 8.2, examples of noisy synthetic X-ray images are shown.

On one hand, the influence of noise on images can be reduced by an increase in the exposure (detector sampling) time due to accumulation. But due to detector saturation, an increased exposure time must be compensated by a decrease in the X-ray intensity, increasing generation and material interaction variance. Averaging multiple images is a common technique. Basically, only Gaussian noise can be reduced significantly by averaging.

## 4. X-ray Measuring Methods and Devices

Compared with widely used Guided Ultrasonic Wave (GUW) measuring methods for damage and defect analysis, X-ray imaging has advantages and disadvantages:GUW signals offer lower spatial resolution (limited by wavelength and material properties) than X-ray images, limiting the analysis to larger defects, especially for 3D reconstruction [35];GUW signals are complex, and signal features related to damage features represent only a small fraction of the entire signal (on time and amplitude scale) and can be ambiguous (indirect feature function), and the signals are composed of different contributions (reflection, diffraction, attenuation, mode conversion);GUW signals are difficult to simulate accurately with respect to real measured signals, moreover in complex, inhomogeneous, and composite/layered materials with a lot of reflections, diffraction, scattering, and mode conversion;X-ray images pose higher spatial resolution limited mainly by the source and detector and bases primarily on direct radiation attenuation;X-ray images can be accurately simulated (i.e., numerically computed) as long as there is an accurate material model and only attenuation along a straight path (ray) is considered;X-ray images directly show the structural distribution of the material based on density variation;X-ray images can are subject to low contrast if the damages or defects are small and/or the density of the defects to be detected are close to the density of the host material.

In this work, three different X-ray measuring device classes are used, summarized in Table 3:Low-Q [36];Mid-Q;High-Q.

All devices can be used to perform single- and radial multi-projection transmission imaging of specimens. X-ray radiography delivers only a two-dimensional integral (along the depth axis) material attenuation map, whereas multi-projection measurements can be used to compute and reconstruct 3D material density distributions (discussed in the next Section 5). The High-Q μ-CT device is only used to create reference data used for the synthetic data generation. The Low-Q and Mid-Q devices are used for capturing X-ray radiography images from real specimens. It is important to point out that the Low-Q device has a significantly more effective higher detector resolution (5 times) than the Mid-Q device.

**Table 3 sensors-24-02933-t003:** Different measuring device classes used in this work.

Feature/Device Class	High-Q	Mid-Q	Low-Q
Single-Projection	Yes	Yes	Yes
Multi-Projection (CT)	Yes	Yes	Maybe
Focal Spot Diameter	5 μm	0.75 mm	∼0.8 mm
High Voltage/Current	80–160 kV/1 mA/50 W	40–180 kV/10 mA/500 W	40–65 kV/1 mA/30 W
Detector	2000 × 2000 CCD 0 μm Scintillator Optics, Imaging Microscope	1000 × 1000 Flat Panel CMOS, 200 μm Direct Scintillator Pixel Coupling	1920 × 1080 CMOS 3/40 μm Screen + Imaging Optics
Digital Resolution [Bits]	16	16	12
Sampling Time	500 ms–10 s	100 ms–1 s	5–10 s
Distance Object/Source	5–10 cm	20–70 cm	10–30 cm
Signal–Noise Ratio (SNR)	High	Mid	Low
Spatial Resolution	High	Low	Mid (!)
Geometrical distortion	Maybe	No	Yes (cushion, defocus)
Approximate Costs	1000 k€ (Zeiss)	500 k€ (Yxlon, IFAM)	1 k€ (Bosse)

## 5. CT Analysis

The CT data were only measured and analyzed for reference purposes, i.e., to obtain typical pore statistics and a reference pore set used for simulation.

The analysis process flow was as follow:Independent auto-cropping of bounding box of material area in all slices using gradient-based edge detection and background-level (black) suppressing.Image intensity homogenization using a horizontal average intensity line profile (integrating pixel intensity perpendicular to the line within a given intensity interval to avoid adulteration by pores and material boundaries).Binarization and feature marking of pores by using a Boolean union operation of output images from a locally adaptive threshold operation (CLAHE) and a global binarization using threshold calculated for each image using Otsu’s method. The adaptive local size was chosen with 20 pixels, and an adaptive compensation parameter was set to 10. The Otsu margin was set to 10.A pixel coordinate set is created from all marked pixels contained in all slice images (3D point cloud volume).The pixel coordinate set is clustered by using DBSCAN with parameters ε = 2 and *minPoints* = 20.The convex 3D hull is computed for each cluster, which should contain points belonging to one pore (or a melted cluster of pores).An ellipsoid fit is applied to the convex hull points, finally delivering the center position, the axis sizes of the ellipsoid, and the angles of the axis vectors with respect to the unity coordinate system axis vectors.

The ellipsoid approximation (least square fit) is solved by an Eigenvalue problem and matrix inversion. The core is a design matrix computed from the coordinates of the convex hull points and the solution of the normal equation:(5)$$\begin{array}{l} D=\left(\begin{array}{c} x^{2}+y^{2}-2z^{2}\\ x^{2}+z^{2}-2y^{2}\\ 2xy\\ 2xz\\ 2yz\\ 2x\\ 2y\\ 2z\\ \vec{1} \end{array}\right)\\ d_{2}=x^{2}+y^{2}+z^{2}\\ \mathrm{solve}\left(uD\cdot D^{T}=d_{2}\right)\to u \end{array}$$
with *u* as an intermediate result for the solution of an Eigenvalue problem with intermediate matrix *A* and vector *v* delivering the ellipsoid axis sizes and axis angles (*R* is computed from *u*):(6)$$\begin{array}{l} v=\left(\begin{array}{c} u_{1}+u_{2}-1\\ u_{1}-2u_{2}-1\\ u_{2}-2u_{1}-1\\ u_{3}\\ \ldots \\ u_{9} \end{array}\right)\\ A=\left(\begin{array}{cccc} v_{1} & v_{4} & v_{5} & v_{7}\\ v_{4} & v_{2} & v_{6} & v_{8}\\ v_{5} & v_{6} & v_{3} & v_{9}\\ v_{7} & v_{8} & v_{9} & v_{10} \end{array}\right)\\ \mathrm{solve}\left(-Ac=\left(\begin{array}{c} v_{7}\\ v_{8}\\ v_{9} \end{array}\right)\right)\to c \end{array}$$
with *c* as the ellipsoid center vector.
(7)$$\begin{array}{l} T=\left(\begin{array}{cccc} 1 & 1 & 1 & c_{1}\\ 1 & 1 & 1 & c_{2}\\ 1 & 1 & 1 & c_{3}\\ 1 & 1 & 1 & 1 \end{array}\right)\\ R=T\cdot A\cdot T^{T}=\left(\begin{array}{cccc} l & l & l & \cdot \\ l & l & l & \cdot \\ l & l & l & \cdot \\ \cdot & \cdot & \cdot & r \end{array}\right)\\ \left\{\hat{e},\vec{e_{v}}\right\}=\mathrm{eigen}\left(\frac{R_{l}}{-R_{r}}\right) \end{array}$$

The axis sizes of the ellipsoid are given by the square root of the absolute value of the Eigenvalues *e_v_*.

The convex hull and ellipsoid approximation is an oversimplification of commonly complex shapes of pores but necessary to create a parameterizable CAD model. A summary of such an analysis from one sample specimen is shown in Table 4, and the average size and volume distributions (for the first 500 pores sorted by size) are shown in Figure 7. Finally, selected views created by ParaView of the entire μ-CT volume are shown in Figure 8, while Figure 9 depicts examples of point clouds representing clustered pores and associated convex hull point cloud approximations. The pores can be seen in the semitransparent and density highlighted view. The contour filter shows a high density of pores, which is not expected by experience and simulation of such a high-pressure die casting process. Therefore, the pore analysis results must be considered with care, although they are used as a reference set for the simulation. There is no alternative method to characterize pores in volumes (micrograph slicing also modifies pores and only small number of slices can be analyzed).

## 6. Image Pre-Processing

### 6.1. Auto-Cropping

Automatic content extraction in images is commonly performed by edge filtering. In this work, the sub-part of an image containing the specimen must be extracted in X-ray radiography and reconstructed CT slice images.

In our work, we used a simple absolute value intensity gradient computation to intensify edges:(8)$$G_{x,y}=\left|I_{x,y-1}+I_{x-1,y}+4I_{x,y}+I_{x,y+1}+I_{x+1,y}\right|$$

We apply some pre- and post-thresholding intensity clippings:(9)$$\begin{array}{l} T_{\left(x,y\right)}^{b}=\{\begin{array}{ccc} 0 & : & I_{x,y}<t_{b}\\ I_{x,y} & : & \mathrm{else} \end{array}\\ T_{\left(x,y\right)}^{b\neg }=\{\begin{array}{ccc} 0 & : & I_{x,y}>t_{b}\\ I_{x,y} & : & \mathrm{else} \end{array}\\ T_{\left(x,y\right)}^{f}=\{\begin{array}{ccc} 0 & : & I_{x,y}<t_{f}\\ 1 & : & \mathrm{else} \end{array}\\ T_{\left(x,y\right)}^{f\neg }=\{\begin{array}{ccc} 0 & : & I_{x,y}>t_{f}\\ 1 & : & \mathrm{else} \end{array}\\ T_{\left(x,y\right)}^{r}=\{\begin{array}{ccc} 0 & : & I_{x,y}<t_{1}\;\;\;\;\mathrm{or}\;\;\;\;I_{x,y}>t_{2}\\ 1 & : & \mathrm{else} \end{array} \end{array}$$

The threshold operations *T^b^* are applied to the input image, the threshold operations *T^f^* and *T^r^* are applied to the gradient-filtered images *G* to create a binarized image. The maximal or largest fitting rectangular bounding box is determined from the binary image finally used for image cropping. There are positive and negative (¬) polarity threshold operations. In radiography images (captured by intensity detectors), the background intensity is higher than the intensity of the matter region, requiring the negative polarity operations; in reconstructed CT slices images, it is vice versa, requiring the positive operations.

### 6.2. Histogram Normalization

Image normalization with respect to low-frequency intensity inhomogeneities is only applied for the CT slice images for further pore feature marking using classical image binarization algorithms. The radiography images as the input for the data-driven feature marking models are used as is.

### 6.3. CLAHE

The Contrast-Limited Adaptive Histogram Equalization (CLAHE, Zuiderveld, 1994) algorithm is a widely used algorithm to homogenize images with complex geometrical intensity gradients aimed at automatically enhancing the contrast of images. A review of CLAHE applications can be found in [37]. Homogenization maximizes the contrast to amplify feature candidates. CLAHE uses histograms to compute over various tile sections of the image. As a result, even in areas that are lighter or darker than the majority of the image, local levels of detail can improve. CLAHE can improve images, overcoming low contrast, noise, and poor structure edges. We use CLAHE as a pre-processing stage only in the CT pore analysis to enhance CT slice images and for the deep learning SAM approach.

It operates iteratively through several key steps:Histogram Computation: Initially, the histogram of pixel intensities within the image is computed. This histogram represents the distribution of intensity values across the image.Adaptive Partitioning: The image is divided into small overlapping tiles or patches. The size of each tile is typically chosen to be small enough to capture local variations in intensity effectively.Histogram Equalization within Tiles: Histogram equalization is independently applied to each tile. This process enhances the contrast within each tile by stretching the intensity range, thereby improving local contrast.Contrast Limiting: To prevent over-amplification of noise in regions with low local contrast, contrast enhancement is limited. This is often achieved by clipping the cumulative histogram within each tile.Interpolation: Finally, the contrast-enhanced tiles are combined to reconstruct the final enhanced image. This step may involve interpolating or blending neighboring tiles to ensure smooth transitions between regions.

### 6.4. Profile-Based Intensity Homogenization

X-ray images pose spatial low-frequency average intensity inhomogeneities due to inhomogeneous X-ray beam illumination (radiography) and due to poly-chromatic beam spectra (CT reconstruction, requiring beam hardening). For the computation of the reference dataset, we used CT slice images and performed pore feature marking with classical binarization techniques. These are sensitive to low-frequency intensity variations that must be equalized before application.

Commonly, there is an intensity gradient parallel to the coordinate system axis system. Therefore, an averaged profile line integral scan is sufficient to correct intensity gradients along one (or two) axis (axes). Thus, if we want to correct an intensity gradient along the *x*-axis, we have to calculate the average intensity in the *y*-direction, ultimately delivering a correction vector.

Care must be taken when computing the integral due to specimen boundaries (only averaging the specimen region) and the pores as features, which must be excluded from the intensity averaging process. This is again performed with a conditional intensity thresholding integral algorithm using only pixel with an intensity within a range [*t*_0_,*t*_1_].
(10)$$\begin{array}{l} A_{x}\left(x\right)=\frac{\sum_{y\forall y|I_{x,y}\in \left[t_{0},t_{1}\right]}I_{x,y}}{\sum_{y\forall y|I_{x,y}\in \left[t_{0},t_{1}\right]}1}\\ C_{x}\left(x\right)=\frac{\max\left(A_{x}\right)}{A_{x}}\\ A_{y}\left(y\right)=\frac{\sum_{x\forall y|I_{x,y}\in \left[t_{0},t_{1}\right]}I_{x,y}}{\sum_{x\forall y|I_{x,y}\in \left[t_{0},t_{1}\right]}1}\\ C_{y}\left(y\right)=\frac{\max\left(A_{y}\right)}{A_{y}} \end{array}$$

The X-ray images *I* are corrected by multiplication with matrix-expanded *C_x/y_* vectors in the *x*- and *y*-directions, respectively.

### 6.5. Thresholding

Thresholding is a fundamental technique in image processing used to separate objects or features of interest from the background based on their pixel intensity values. It works by setting a global or local threshold value, which is a predefined intensity level. Pixels with intensity values above the threshold (or within an interval) are classified as foreground (object), while those below are classified as background. Thresholding typically requires image preprocessing prior to its application. This includes spatial intensity homogenization as discussed before, denoising of the image (e.g., by using Gaussian filters), and application of CLAHE.

Zack et al. [38] introduced the triangle thresholding method. It involves three pivotal steps:Normalizing the intensity histogram’s height and dynamic range.Identifying point A on the intensity axis that is a crossing point of a line perpendicular to a line connecting the maximum and right-hand minimum of the intensity distribution while the distance between the line and the histogram is maximized.Integrating a fixed offset.Finally, the optimal threshold (T) is determined, with a slight fixed offset relative to the average brightness of the background within a six-pixel radius of the object [39].

Otsu’s thresholding [40] as well as adaptive thresholding [41] methods are alternative candidates. Combining them by Boolean union operations can suppress noise and wrong feature marking significantly, as shown in the next Section.

### 6.6. Examples

In the following Figure 10, the different image processing algorithms are demonstrated for a specific CT slice image (from μ-CT measurement of one of the AluDC specimens). The intensity homogenization using the profile scan correction is superior compared to histogram-based homogenization and a prerequisite for further binarization algorithms. The Otsu algorithm computed an optimal global threshold, whereas the adaptive thresholding algorithm operates locally. The Boolean union combination produces the most accurate and stable pore binarization results. The CLAHE algorithm provides no suitable enhancement of the CT slice image for binarization (over-amplification of contrast).

## 7. Feature Marking Models

There are basically two classes of feature detectors:Model-driven using geometrical, statistical, and numerical algorithms [42];Data-driven detectors learning approximated mapping functions from annotated training data, e.g., by using CNN.

Classical numerical algorithms using image binarization for feature marking typically require a sufficient and distinct contrast of the features with respect to the background (sufficient SNR). This work entails the defects producing intensity variations in radiography images with a weak contrast. In doing so, it will be demonstrated that data-driven approaches are better suited for low-contrast feature marking applications.

### 7.1. Semantic Pixel Classifier CNN

The main objective of this work is the deployment of an automated feature detector applied to single-projection X-ray radiography images delivered by a Low-Q (low-cost) and Mid-Q X-ray instrument to detect hidden defects in materials, here specifically pores in high-pressure die-cast aluminum plates.

The input is an X-ray radiography image (inverse relation of intensity to material density), the output is a feature map image that marks pores by using a binary classifier with the classes *N* (background) and *P* (pore), as shown in Figure 11. Finally, a post-analysis provides the geometric parameters and positions of the defects.

A pixel classifier is commonly implemented with a Convolutional Neural Network (CNN), mostly with only one or two convolution–pooling layer pairs (CPL). The input of the CNN is a sub-window masked out from the input image at a specific center position (*x*,*y*). The output is a class (or a continuous score value in the range [0, 1] as an indicator level for a class) related to the center pixel of the segment. The neighboring pixels determine the classification result. The window (image segment) is moved over the entire input image, producing the respective feature output image. Each output pixel computation requires one application of the CNN to an image segment.

Typical architecture parameters of such a pixel classifier are shown in Table 5.

The segment input size can vary. Smaller segments tend to false-classify pixel of larger shapes (multiple times larger than the segment size), and larger segments tend to false-classify background pixel. Note that *N* = ¬*P* means that the background class is anything else (here, mostly X-ray image noise). The segment size must be adapted to the average size of shapes to be detected. The number of filters per convolution layer can typically vary between 4 and 8. A second convolution–pooling layer pair is optional and commonly does not improve prediction accuracy. The pooling layer commonly uses a maximum pooling function. The activation function applied after the convolution is commonly a linear or cut-off linear function. Nonlinear functions like sigmoid do not improve classification results and are harder to train.

The computational complexity of the entire feature marking process for one image consisting of *N* pixels is Θ(*N*). A CNN with the above given parameters and one CPL pair requires about 80,000 (Conv) + 3200 (Relu) + 1600 (Pool) + 1600 (FC) = 86,400 arithmetic unit operations. Because all segment CNN operations are independent, the computation can be massively parallelized using Cellular Automata (CA) or Graphics Processing Units (GPU). In [43], we showed that such a CNN can be processed by a low-precision arithmetic processor still preserving the accuracy. So, a direct hardware implementation of a pixel classifier with very low overall computation times is possible.

### 7.2. Deep Learning Segmentation with SAM

In this work, we focus on lightweight and small models, which can be trained with a rather small dataset. But we want to compare this model class with established and widely used so-called “Deep Learning” (DL) models, which typically require a large set of data for training, but which are suitable for transfer learning. In contrast to the simple semantic pixel classifier model, the Segment Anything Model (SAM) is a highly complex model. Details can be found, e.g., in [8,44].

Developed by Meta AI, SAM is an advanced image segmentation model trained on an extensive dataset (SA-1B) featuring more than 1 billion segmentation masks spread across 11 million images. It is specifically designed to interpret human prompts, accepting points, bounding boxes, or text descriptions for segmentation. It was developed to combine prompt able segmentation having real-time performance capabilities by taking inspiration from NLP. SAM’s design lets it adapt to new tasks and image types, thanks to something called zero-shot transfer learning. It gives us high flexibility and adaptability in analyzing images.

Key Features of the Segment Anything Model (SAM) [44]:Zero-shot generalization: SAM can be used to segment objects that it has never seen before, without the need for additional training.Promptable Segmentation: SAM is structured for the promptable segmentation tasks at its core, enabling it to produce valid segmentation masks based on prompts such as points, boxes, and text descriptions.Real-time mask computation: SAM can generate masks for objects in real time. This makes SAM ideal for applications where it is necessary to segment objects quickly, such as autonomous driving and robotics.Zero-Shot Performance: SAM has exceptional zero-shot performance capabilities which are suitable for various segmentation tasks, making it a versatile tool for diverse applications with minimal requirement for prompt engineering.Ambiguity awareness: SAM is aware of the ambiguity of objects in images. This means that SAM can generate masks for objects even when they are partially occluded or overlapping with other objects.

SAM consists basically of three components: An image encoder, a flexible prompt encoder, and a fast mask decoder. The architecture is summarized in Figure 12.

The components process an input image as follows:
Image Encoding:○SAM initiates by encoding the input image into a high-dimensional vector using a vision transformer (currently three models, namely ViT-H, ViT-L, and ViT-B, are available) model.○ViT-H, a large language model, is pre-trained on a massive dataset of images.Prompt Encoding:○The input prompt undergoes separate encoding into a vector representation.○A simple text encoder converts the prompt into a meaningful vector.Combining Representations:○The vector representations from the image and prompt encoding are combined.○These combined vectors encapsulate information about both image content and specified prompts.Mask Decoding:○The combined vector is then passed to a mask decoder, a lightweight transformer model predicting the object mask.Output:○The output of the mask decoder is the predicted mask for the object specified by the input prompt.

The SAM training pipeline involves:
Training image generation: Synthetic image creation through simulation software, annotation, and mask generation;Image preprocessing: Denoising of the images, application of CLAHE (see Section 6), and intensity homogenization;The training images and corresponding ground truth feature masks are divided into patches of size 256 × 256 pixels, and bounding boxes around pores in ground truth masks are created.

For the pore segmentation task, the SAM was fine-tuned on the synthetic X-ray images having annotated pores. The real-world images do not have any ground truth (marked pores) as we are not sure where the pores are exactly located. These images are used for inference and the performance is judged based on the visual inspection.

Due to the static input size of the SAM model, a static segmentation of large images into smaller segments (non-overlapping) is commonly required. As shown in Section 8 that follows, this can result in chess patterns, i.e., neighboring segments show totally different marking results with respect to noise and artifacts.

## 8. Experiments and Results

In this section, the CNN-based pixel classifier and the SAM model are trained with synthetic data derived from the CAD-based X-ray image simulation. The models are tested with the synthetic data to determine the ground truth error, finally applying the models to real image data measured by the Low-Q and Mid-Q X-ray radiography devices. In Figure 13, an example of a training image and its mask annotation are shown.

The feature marking models are used for a pore analysis, eventually providing pore statistics and pore size distributions. The CNN model and SAM are both trained and applied to synthetic and real X-ray images. The SAM model used for the training is ViT-B (sam-vit-base), which is a lighter and faster version as compared to the other two available options (ViT-H, ViT-L). A total of 9 synthetic training images of 398 × 996 pixels resolution were used with patch size of 256 × 256 pixels, step size of 32, and batch size of 2. The optimizer used is an Adam optimizer with the learning rate of 10^−5^. The training time was around 180 min.

Before we present results from the image-based pore analysis workflow, results from a die casting simulation are presented. The aim is to show firstly the limits of widely used die casting simulations to obtain porosity statistics and predictions of porosity, eventually concluding that the experimental analysis using X-ray radiography is still needed to evaluate a die casting manufacturing process in more detail, and optimize it based on this analysis. Secondly, despite the limitations of the process simulation, we use some of its qualitative results in the structural modeling of defects and their geometrical distribution, which depends on the casting process parameters and specimen shape.

### 8.1. Die Casting Simulation

The complete virtual shot is visible in Figure 14. For the present experiments, only the rectangular bending test samples with dimensions of 150 mm × 40 mm × 3 mm originating from the cavity on the left-hand side were considered. The total shot weight was 600 g, the plunger velocity 5 m/s, leading to a much higher velocity at the in-gates. The die temperature was set to a nominal value of 200 °C, which was cross-checked via contact measurements on the die surface using thermocouples: These revealed a slight surface temperature gradient, extending from 200 °C in the lower and 190 °C in the upper part of the relevant cavity.

A parallel casting simulation reflecting the actual production parameters was performed in order to gain insights about the local distribution of different types of porosity to be expected. The simulation model was set up using MAGMASOFT^®^ casting simulation software version 6.0.0.2. The respective data can be matched with pore localization and shape information gained via ADR approaches for verification and validation of the latter.

The geometry of the casting is shown in Figure 14, which depicts the die cavity including the casting system halfway through the filling process. The experiments described below were focused on the bending test sample, i.e., the rectangular plate geometry seen on the left-hand side of the image.

Figure 15 depicts the sequence of die filling as visualized using the tracer particle feature offered by MAGMASOFT^®^, which provides graphical information about direction and velocity of the melt flow, while additionally representing melt age via color coding. The simulation clearly shows that due to the fact that the in-gate does not extend over the full width of the bending test specimen, an “eddy water”-like region forms along both longitudinal edges of this part, while its center is characterized by a constant stream of melt throughout the filling process. This already indicates some probability of entrapped air in the former regions on both sides of the casting.

This understanding is basically confirmed by Figure 16, which illustrates the filling process. Figure 16a on the left reflects the melt flow velocity shortly before the end of the filling phase. Once again, the difference between central and outer regions of the part is clearly visible. Figure 16b provides an indication of air entrapped within the casting and the overflows, underlining once again that the flow pattern leads to a higher risk of gas porosity close to the edges of the part.

In contrast, the general porosity criterion offered by the software, which is meant to capture shrinkage porosity, finds the latter almost exclusively in overflows and thus outside the actual casting (see Figure 17a), which predicts next to no porosity within the part volume). However, the Hot Spot FS Time criterion, which highlights areas which are cut off from feeding prior to complete solidification, indicates the expected possibility of some levels of shrinkage porosity in the center of the plate (Figure 17b).

### 8.2. Feature Marking Using CNN Pixel Classifier

#### 8.2.1. Training

The synthetic image set created by the previously introduced simulation workflow consists of 10 X-ray radiography images with a resolution of 100 μm pixel size and about 400 × 1000 pixels (auto-cropped). Each synthetic image contains 100 randomly but parametrically constrained pores with ground truth labeling.

The pixel CNN classifier was initially trained with a class-balanced 50/50% training set of 10,000 randomly selected segments (20 × 20 pixel size) and 10 epochs. Examples are shown in Figure 18. A default “adadelta” error back-propagation algorithm with a *l2decay* = 0.001 parameter setting was used. The segments were chosen from the pore ROI list. For the pore class (*p*), a segment was created around a center point contained in a pore ROI. For the background class (*n*), a segment was created around a center point from the manually created background ROIs. The SAM model was trained using the synthetic images and the ground truth ROI annotations (poly-lines), too. The SAM model does not require a fine-grained segmentation, but a specific SAM model has a fixed image input size. Therefore, the input image must be segmented with respect to the input size of the SAM model and applying the model to each segment independently.

#### 8.2.2. Synthetic Images

Some selected results of the feature marking using the CNN pixel classifier and applied to synthetic images are shown in Figure 19. The images show the output score of the CNN and the ground truth labels (poly-lines). The CNN was initially trained with a class-balanced training set of 10,000 segments and 10 epochs. There are a lot of false-positive predictions, as can be seen clearly in the feature map images. But by applying an increasing threshold, this FP noise is significantly reduced. A post-training with a negative class (background) biased dataset reduces noise, too.

Assuming a score threshold of 0.9, the false-positive (FP) rate is (for the initially trained model) below 1% of all pixels, and the false-negative (FN) rate is below 1% of all pores.

#### 8.2.3. Real Images

Some selected results of the feature marking using the CNN pixel classifier applied to real radiography images from the Mid-Q device are shown in Figure 20. To test noise sensitivity, X-ray radiography images from rolled aluminum plates (same thickness) were measured, and the feature marking model was applied to the images, too. Selected results are shown in Figure 21. With a score threshold of 0.9, the FP rate is 0.

The Mid-Q device provides a detector resolution of 200 μm pixel size with low noise output (experimental set-up with magnification *M* = 1.6, i.e., resulting theoretical resolution is 125 μm per pixel). The alternative Low-Q device suffers from much higher noise and optical distortions but provides a higher effective resolution of about 45 μm pixel size (with *M* = 1). Tests showed a real resolution limit of the Low-Q device of about 10 LP/mm. The results of the feature marking are shown in Figure 22. It seems that the pixel classifier is much more noise-sensitive (higher FP rate) if applied to the Low-Q images, but this is just a subjective observation. The following statistical pore analysis gives some more results and a comparison. Again, the noise reduces by increasing the score threshold. But a post-training of the CNN model with a background class biased training dataset (again 10,000 segments randomly sampled from the input images) and 20 training epochs results in a decrease in noise without any threshold comparable to a high threshold in the previous model. But the increased background noise immunity results in a lower feature marking rate in the low-noise Mid-Q radiography images.

The results from HPDC simulation predict a specific geometric distribution of pores and different classes of pores with respect to the size distribution. The feature marking of the real images is roughly in accordance with the simulation prediction.

To compare the simple CNN-based pixel classifier with a highly complex and deep neural network model, the same input images were processed by the SAM model. The SAM model was trained with the same synthetic X-ray images. Selected results of SAM feature marking applied to Mid-Q X-ray images are shown in Figure 23. The results differ strongly from the results of the CNN, as shown in Figure 20. The noise level is much higher, dominant pores directly visible in the X-ray images are not well marked, and there are artificial patterns (artifacts). The patterns are characterized by missing pixel markings. This is critical due to the following point clustering-based analysis (see Section 8.3). The gaps can result in a split of sub-clusters and small fake pores.

### 8.3. Pore Analysis

A statistical pore analysis from real images marked with the CNN pixel classifier and the deep SAM model is shown in Figure 24, finally clustered using DBSCAN and fitted to a geometric ellipse model. The pore distributions differ significantly in images from the Mid-Q (low noise, lower resolution) and the Low-Q devices (higher noise and higher resolution). The device with the higher resolution (pixel size 45 μm compared with 200 μm/125 μm) shows a shift towards smaller pores, in alignment to the pore area distribution from the 3D μ-CT analysis (see Figure 7 for comparison). Feature marking in images from the lower resolution Mid-Q device shows a much broader distribution. Comparing feature marking in Mid-Q device images using the pixel classifier (PXL-CNN8, one convolution layer with 8 filters) with the deep SAM model shows similar results, but with respect to the SAM results shifted towards smaller pores. But SAM produces artificial patterns that can be misclustered and result in smaller (dissected) pores.

## 9. Outlook: Detection of Hidden Complex Damages in Composites and Laminates

This section outlines the next level of synthetic data generation of complex structures and defects. Since it is a work under progress, it is considered as an outlook. The previous sections addressed porosity analysis in homogeneous materials by using a data-driven predictor model trained with synthetic data basing on structural modeling. The generation of synthetic CAD models containing the host material and the embedded pores is quite easy. It is just a material-subtractive operation to create “holes” inside the material not being constrained by a constant mass–volume. These defects are introduced at manufacturing time. In contrast, post-manufacturing defects (damages) are much harder to model accurately, e.g., caused by an impact event. Any post-damaging must preserve the constant mass–volume constraint, i.e., a deformation and cracking may not add or remove material. Especially, cracks are difficult to model using CSG, although polygon line chain extrusion can be used to create free-form shapes.

Considering Fiber–Metal Laminates (FML), which consist of metal and glass (or carbon) fiber–resin layers (PREG), impact events are the most important damage cause. An impact event can create:Deformations;Cracks;Delaminations.

All these material modifications must be geometrically modeled to generate realistic X-ray radiography and CT projection images. Basically, three CSG modeling approaches can be used, as shown in Figure 25:Boolean union and difference computation of damaged parts with an undamaged solid material cube (aluminum and PREG both!), 3D extrusion of half-boundary profiles based on CT image analysis and hull boundary poly-lines → Modeling of symmetric damages only;3D extrusion of half-convex hull of each layer + union extension of non-damaged plate → symmetric damages only;Point cloud convex hull solid creation (aluminum and PREG); no extrusion → Modeling of asymmetric damages possible.

**Figure 25 sensors-24-02933-f025:**
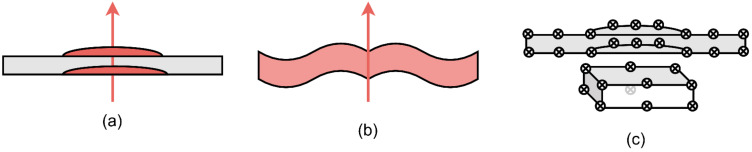
CSG-CAD modeling of deformations: (**a**) Hybrid with cubes and extruded half-boundary profiles; (**b**) Half-convex hull extrusion; (**c**) Point cloud convex hull extrusion.

The first experiment was performed using boundary layer poly-lines from real CT slice images (μ-CT device) created by a semi-automatic boundary tracking algorithm. The specimen under test was an aluminum–PREG multi-layer plate. Actually, only the deformations of the aluminum and PREG layers were considered. The fiber PREG layer was modeled as a solid material. Finally, a multi-material STL mesh-grid model was created and used for X-ray radiography and multi-radial projections (for CT reconstruction). The plate is shown in Figure 26.

Results of the CT reconstruction of simulated radial projections (800, full turn) using a sine-wave filtered back-projection (FBP) algorithm are shown in Figure 27 for CAD modeling *A* and *B* approaches. The CT reconstruction showed accurate agreement with real CT data. To test the constant mass–volume constraint, horizontal intensity profiles were computed from the X-ray radiography image. First, a radiography image of the aluminum without PREG material was created and analyzed. Due to the deformation, a slight increase in the intensity in the center was expected and observed in both CAD modeling approaches (and real radiography images). The boundaries of the deformed metal layers could be accurately recorded from the real data. The analysis of radiography images with the PREG layers only shows a significant decrease in the center intensity, which clearly identifies a violation of the constant mass–volume constraint for the PREG layers. The boundaries of the PREG layers are fuzzy and hard to discriminate in the real CT images.

## 10. Conclusions

Porosity characterization and analysis of die-cast parts and materials is of high relevance, specifically in view of recent developments in the industry like Gigacasting. Our study once more highlights the fact that widely used model-based physical simulation of die casting processes neither provides sufficiently detailed quantitative nor qualitative results of statistical and geometrical pore distributions inside the materials. Its capabilities in terms of predicting local material properties must therefore be limited, and experimental analysis is mandatory. Nevertheless, results and fundamental analyses based on these process simulations can be used for synthetic data generation, as introduced in this work. Using high-resolution μ-CT scans for this purpose is expensive and time-consuming. X-ray radiography could be an economic alternative, but the characteristically low contrast and lowered Signal-to-Noise ratio of defect features is a major obstacle to its exclusive application in this context, the more so since the classical image processing solutions often prove unsuitable for sufficiently detailed pore characterization and classification. Therefore, data-driven feature marking models are gaining interest. These models require a ground truth dataset with sufficient parameter variance that is currently not available. As a working alternative, synthetic data generation based on CAD modeling and X-ray simulation can be employed. This approach has been demonstrated in the present study and qualitatively validated against conventional casting simulation results.

The pixel classifier CNN model is a local image operator and is independent from the size of the source image and can be applied to any image sizes. The application requires a temporary dynamic and overlapping segmentation of the input image (mask window). To speed up the feature marking process, a striding larger one can be chosen. The SAM model is a global image operator with a static input size requiring static non-overlapping image segmentation.

The CNN pixel classifier trained with synthetic images showed convincing results with respect to false-positive and -negative prediction errors. The application of the trained model to the original ground truth synthetic images showed an accuracy of 0.99 and false-positive rate relative to all image pixels below 1%. The low-complexity CNN pixel classifier as well as a highly complex deep learning segmentation model create binarized images from X-ray radiography images. A point clustering of all marked pixels showed different statistical pore analysis results. Using images from a Mid-Q device (mid-resolution, low noise), the pore area distribution is skewed towards large pores compared with the reference set from the μ-CT analysis. There is no significant difference between the pore area distributions retrieved from CNN and SAM binarized images, but SAM binarized images showed artificial patterns (artifacts) which resulted in the splitting of larger pores into smaller clusters by the clustering algorithm. Using a Low-Q device (featuring higher noise and optical distortions, but higher resolution), the pore distribution agrees better with the one describing the reference set. The results from HPDC simulation predict a specific geometric distribution of pores and different classes of pores with respect to the size distribution. The feature marking of the real images and the following pore analysis is roughly in accordance with the simulation prediction.

To conclude, the simple semantic pixel classifier is suitable for a statistical porosity analysis of die-cast materials and outperforms a common deep learning model like SAM. The missing ground truth and data annotations in experimental data were overcome using a unified simulation workflow. The simulation includes CAD modeling of materials and defects with CSG and Monte Carlo methods, and X-ray image computation based on the CAD models using the Beer–Lambert law. The simulated synthetic data were used to train the ML model, which was finally applied to experimental data showing satisfying results by comparing the statistical pore size distributions with the reference set from μ-CT.

## Figures and Tables

**Figure 3 sensors-24-02933-f003:**
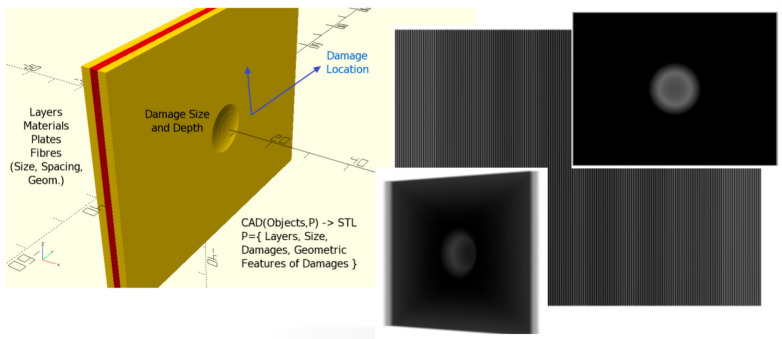
(**Left**) Complex FML model with single-fiber modeling and a simplified impact damage, 10,000 fibers with 150 μm diameter. Damage size and location can be changed. (**Right**) Computed X-ray images: Frontal and 45° projections, detector pixel size 100 μm, with and without impact damage.

**Figure 4 sensors-24-02933-f004:**
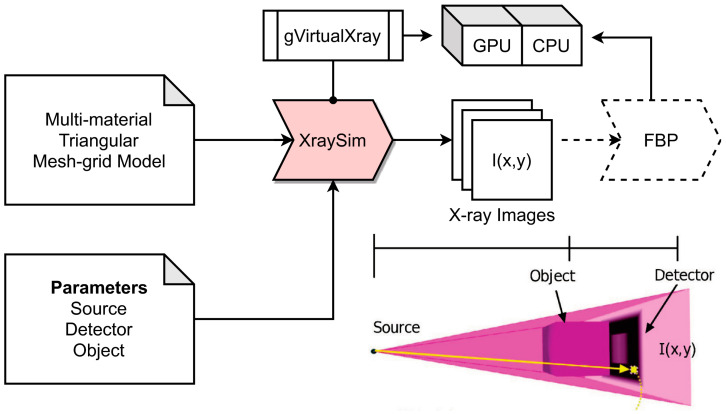
Details of the X-ray simulation flow using GPU-based image computation (gVirtualXray). The Filtered Back-Projection (FBP) is optional and only used for CT simulation.

**Figure 5 sensors-24-02933-f005:**
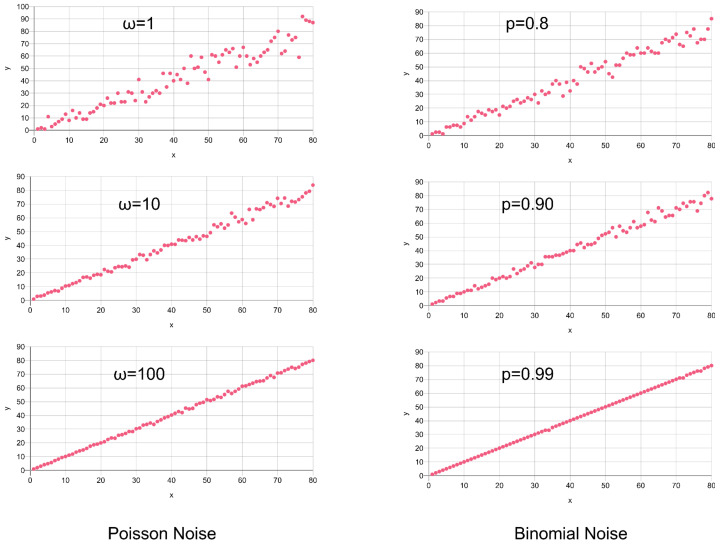
Post-adding of noise to X-ray images (normalized intensity [0, 80]) using Poisson and binomial distributions showing an increasing Signal-to-Noise ratio (SNR) with respect to low absolute signal values converging to a constant SNR for higher values.

**Figure 6 sensors-24-02933-f006:**
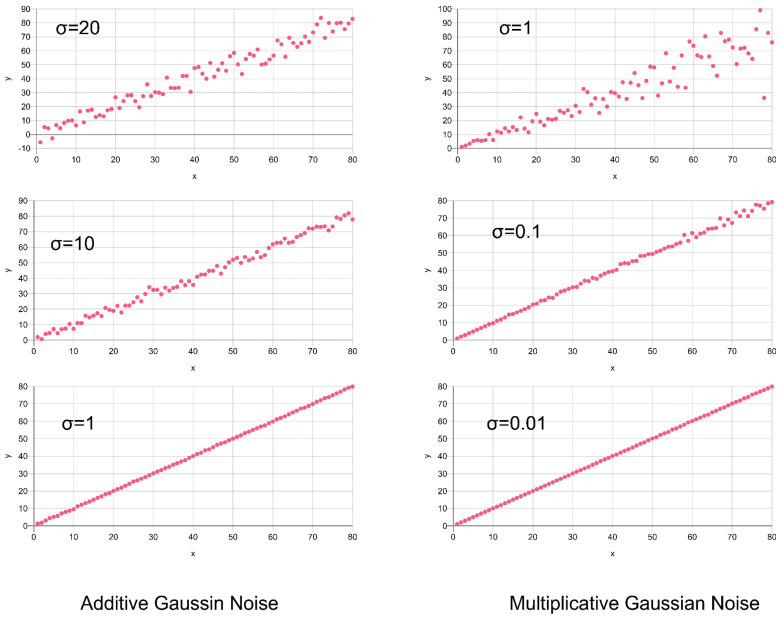
Post-adding of Gaussian noise to X-ray images (normalized intensity [0, 80]) showing a continuously increasing Signal-to-Noise ratio (SNR) with higher signal values.

**Figure 7 sensors-24-02933-f007:**
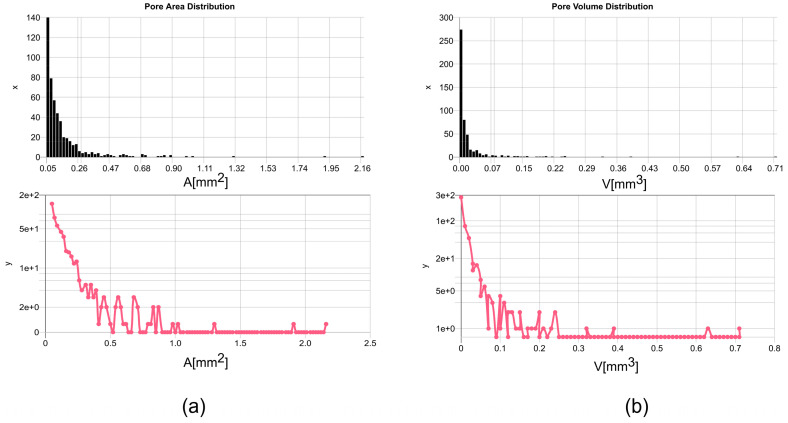
Statistical distribution of the pore ellipsoid area (**a**) and volume (**b**) for the first 500 largest pores shown with linear and logarithmic density axis.

**Figure 8 sensors-24-02933-f008:**
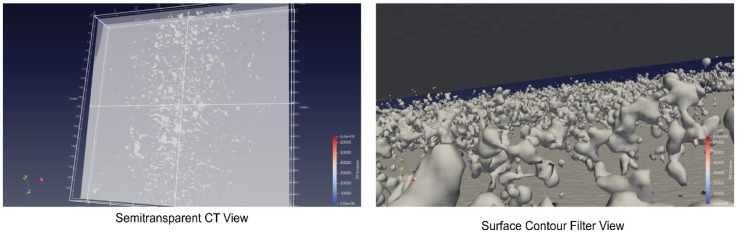
μ-CT volume rendering with semitransparent view and contour fit (using ParaView).

**Figure 9 sensors-24-02933-f009:**
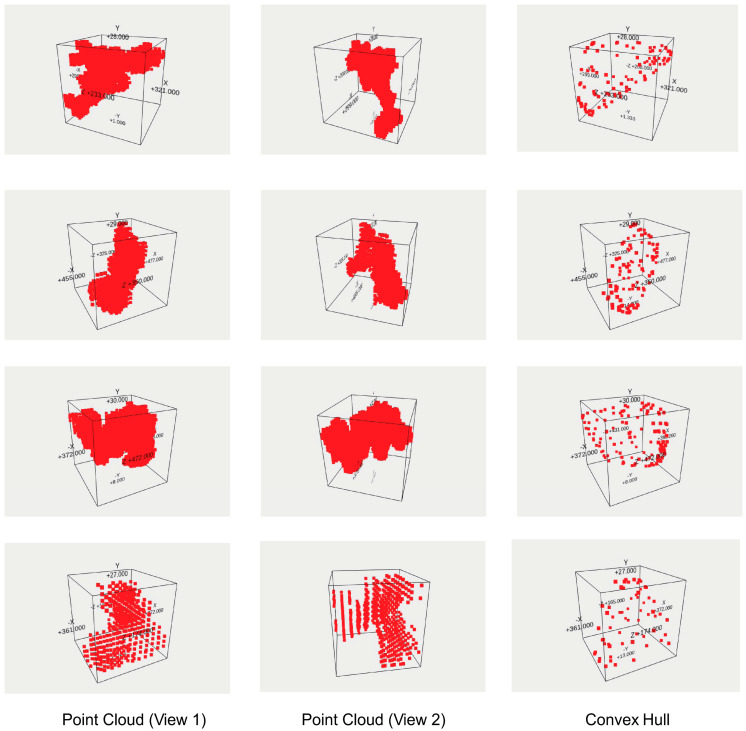
Different clustered pores (**left** and **middle**) and the computed convex hull point cloud (**right**). From top to bottom decreasing pore sizes.

**Figure 10 sensors-24-02933-f010:**
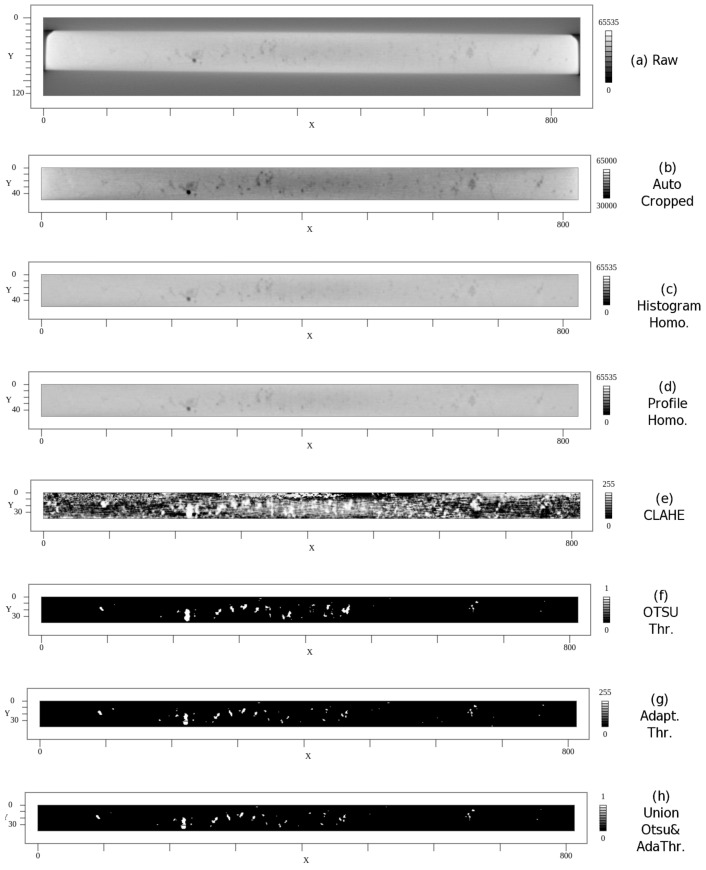
Different image processing algorithms applied to a raw (original) CT slice image (output from CT reconstruction and filtering algorithms). The last image shows the binary combination of the Otsu and adaptive threshold computations, finally used to extract pore features.

**Figure 11 sensors-24-02933-f011:**
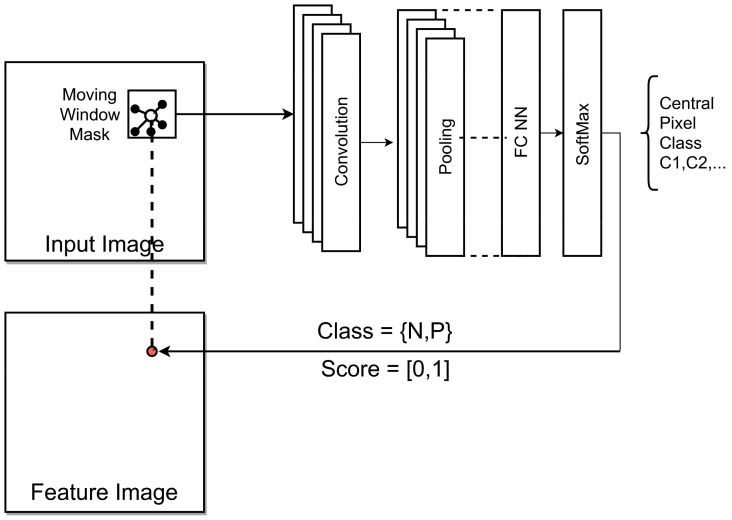
Architecture and data flow of the semantic pixel classifier using a flat CNN (FC-NN: Fully connected neural network).

**Figure 12 sensors-24-02933-f012:**
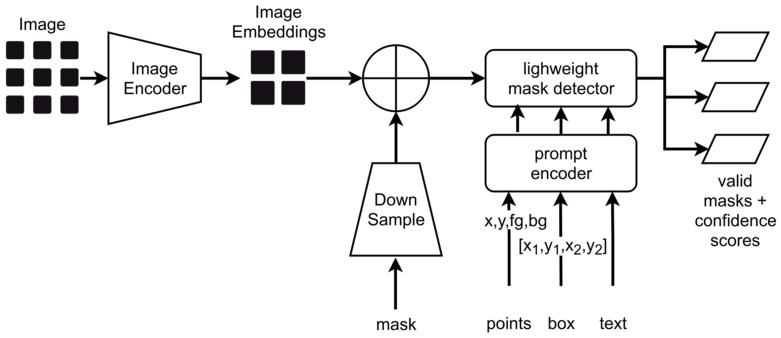
Basic architecture of SAM model (adapted from [44]).

**Figure 13 sensors-24-02933-f013:**
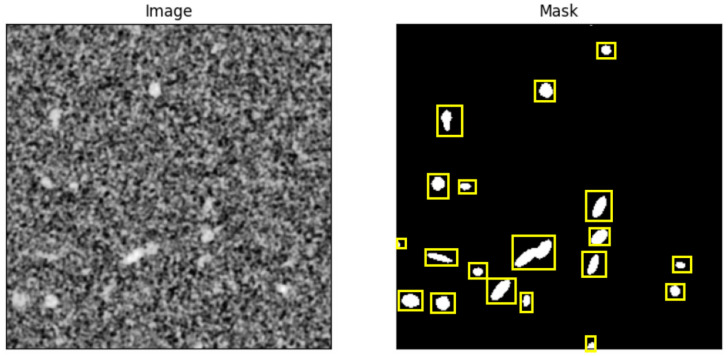
(**Left**) Example of a synthetic X-ray training image. (**Right**) Rectangular ROI pore mask annotations from CAD model.

**Figure 14 sensors-24-02933-f014:**
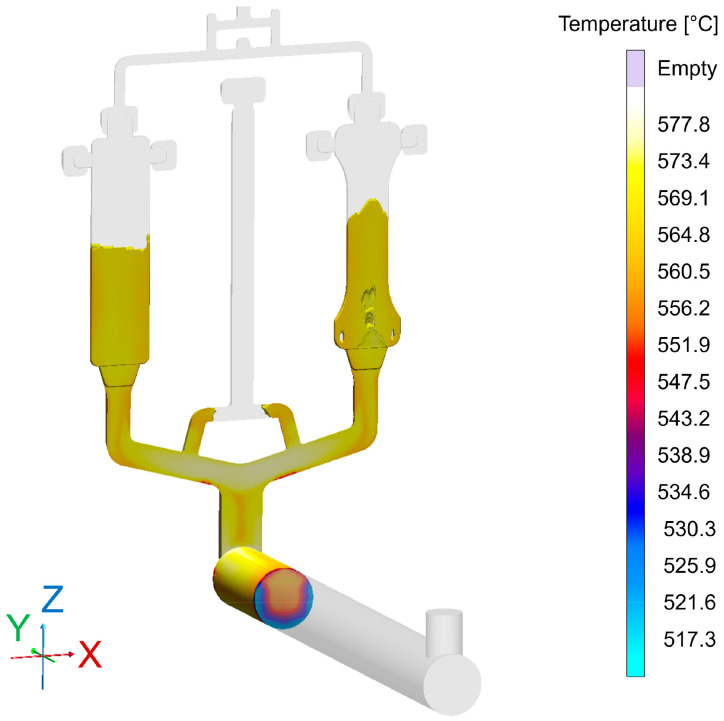
Die casting simulation: Geometry of the casting—die cavity including shot chamber, runner, overflows, etc. during mold filling. For the experiments, the rectangular bending test sample on the left was used. The color plot shows the temperature distribution at the beginning of the filling process.

**Figure 15 sensors-24-02933-f015:**
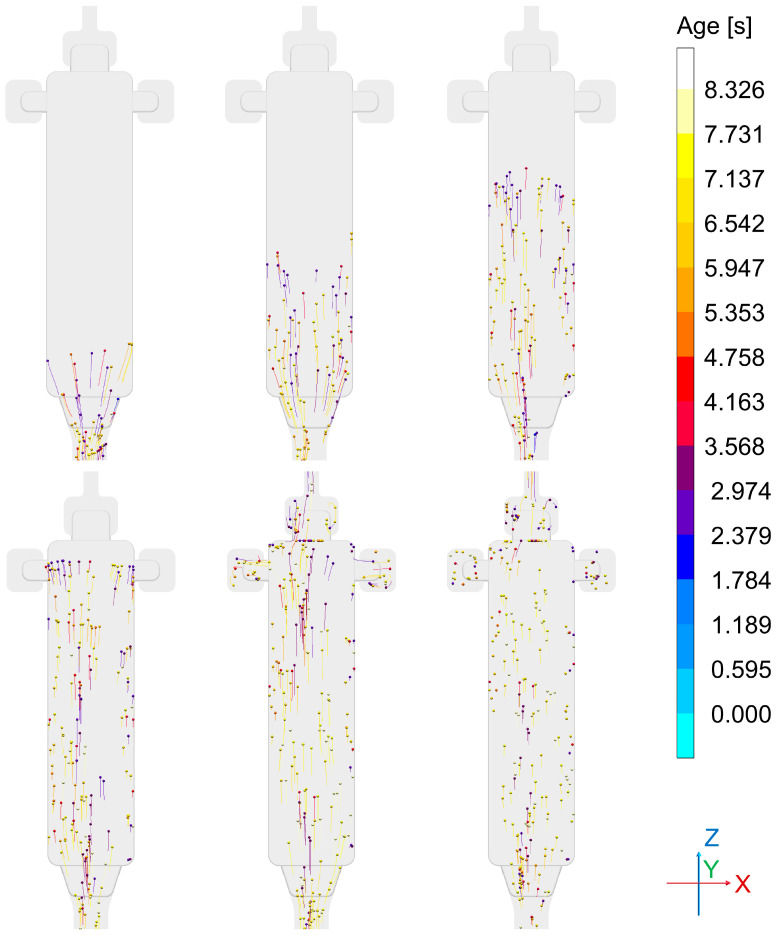
Die casting simulation: Sequence of die filling. Color coding denotes the age of the respective melt volume, while arrows indicate the direction and velocity of melt flow.

**Figure 16 sensors-24-02933-f016:**
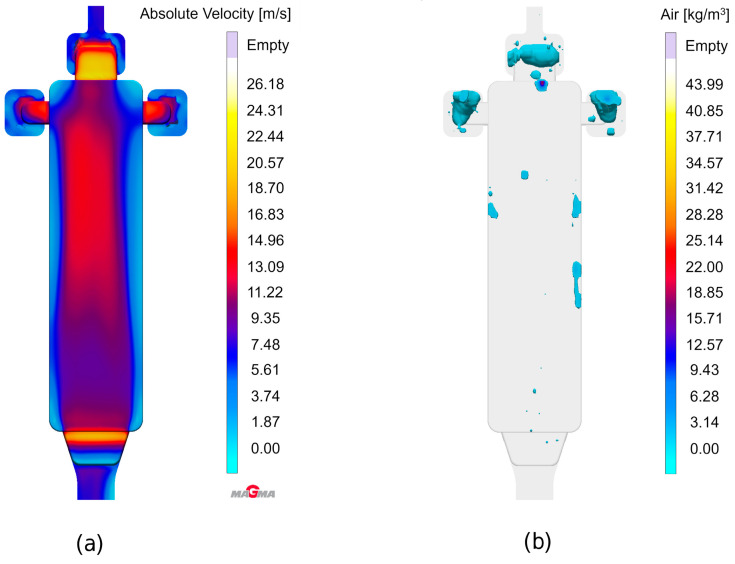
Die casting simulation representing the filling process: (**a**) Melt velocity distribution shortly before the end of the filling phase, (**b**) entrapped air mass prediction broadly reflecting the flow pattern.

**Figure 17 sensors-24-02933-f017:**
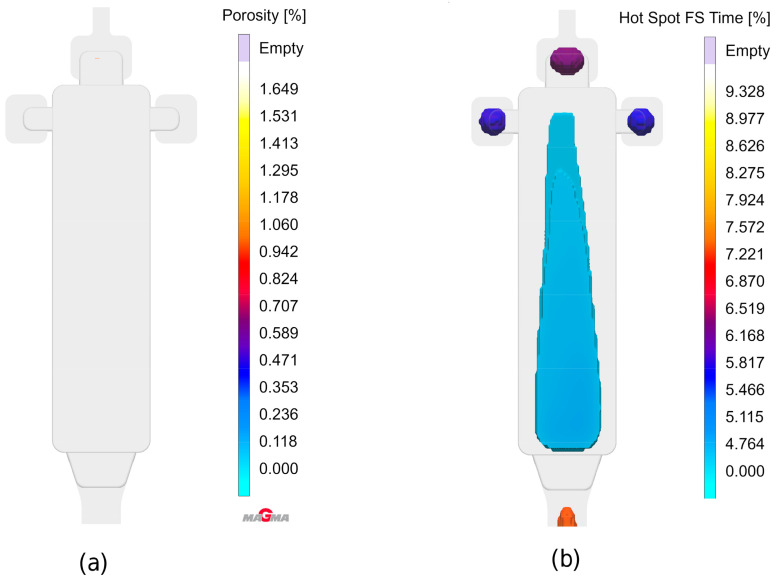
Die casting simulation depicting the general porosity prediction: (**a**) Prediction of porosity; (**b**) Hot spot FS time criterion indicating risk of occurrence of shrinkage porosity.

**Figure 18 sensors-24-02933-f018:**
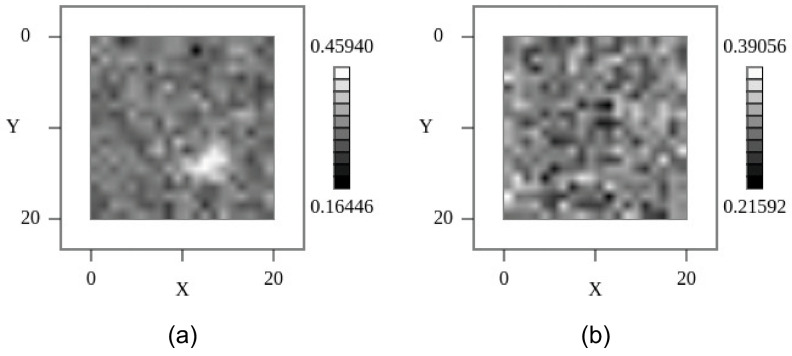
Two example segments (20 × 20 pixels) extracted from the synthetic X-ray images with 20% noise level: (**a**) With pore; (**b**) Without pore.

**Figure 19 sensors-24-02933-f019:**
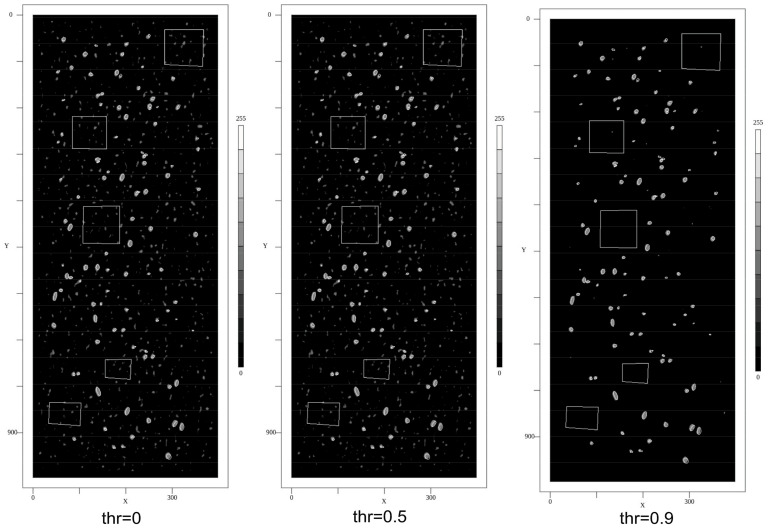
Feature map images (pore marking) computed by the CNN pixel classifier using the synthetic X-ray radiography images as input for three different output thresholds (score of the classifier [0, 1]). The ground truth ROI polygons are shown as an overlay, too.

**Figure 20 sensors-24-02933-f020:**
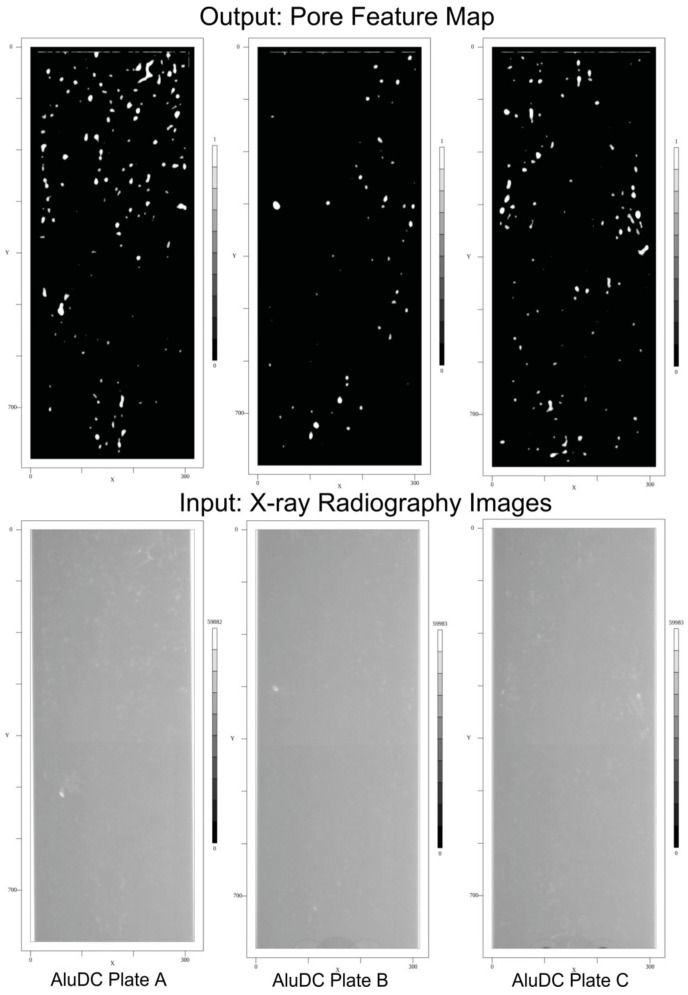
Feature map images (pore marking) computed by the CNN pixel classifier and measured X-ray radiography images (Mid-Q) as input (for three different plates).

**Figure 21 sensors-24-02933-f021:**
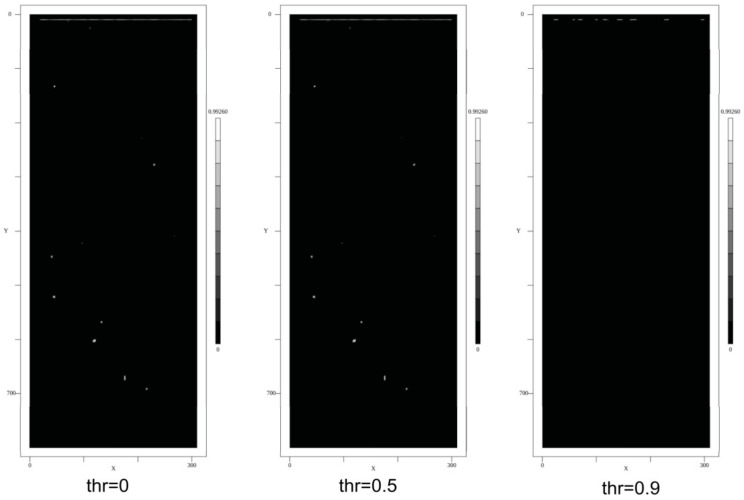
Feature map images of rolled aluminum plates without pores computed by the CNN pixel classifier for different score thresholds (Mid-Q). Expected result: Black without feature marking!

**Figure 22 sensors-24-02933-f022:**
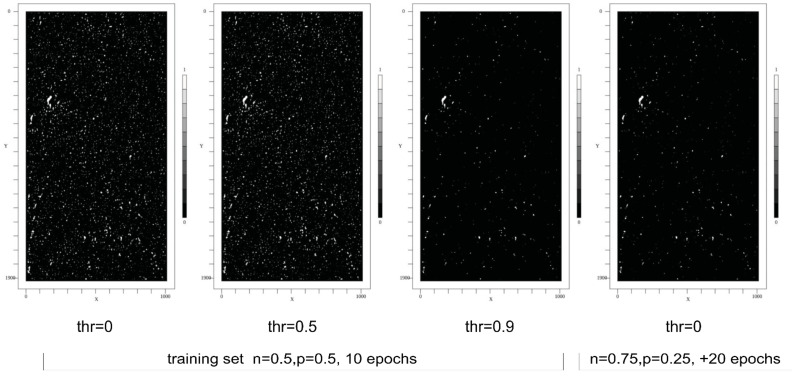
Feature map images (pore marking) computed by the CNN pixel classifier and measured X-ray radiography images (Low-Q) as input for different score thresholds (**Left**); CNN model trained with balanced n/p set; (**Right**) Post-training with biased set n = 75%, *p* = 25%.

**Figure 23 sensors-24-02933-f023:**
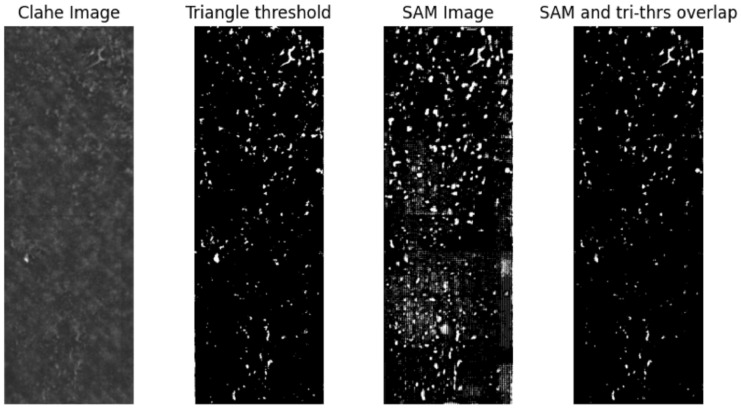
Pore segmentation results obtained from the SAM, triangle thresholding and the overlapping results from both methods. The CLAHE image is obtained by first denoising, then removing uneven illumination, and then applying CLAHE (applied to Mid-Q X-ray images).

**Figure 24 sensors-24-02933-f024:**
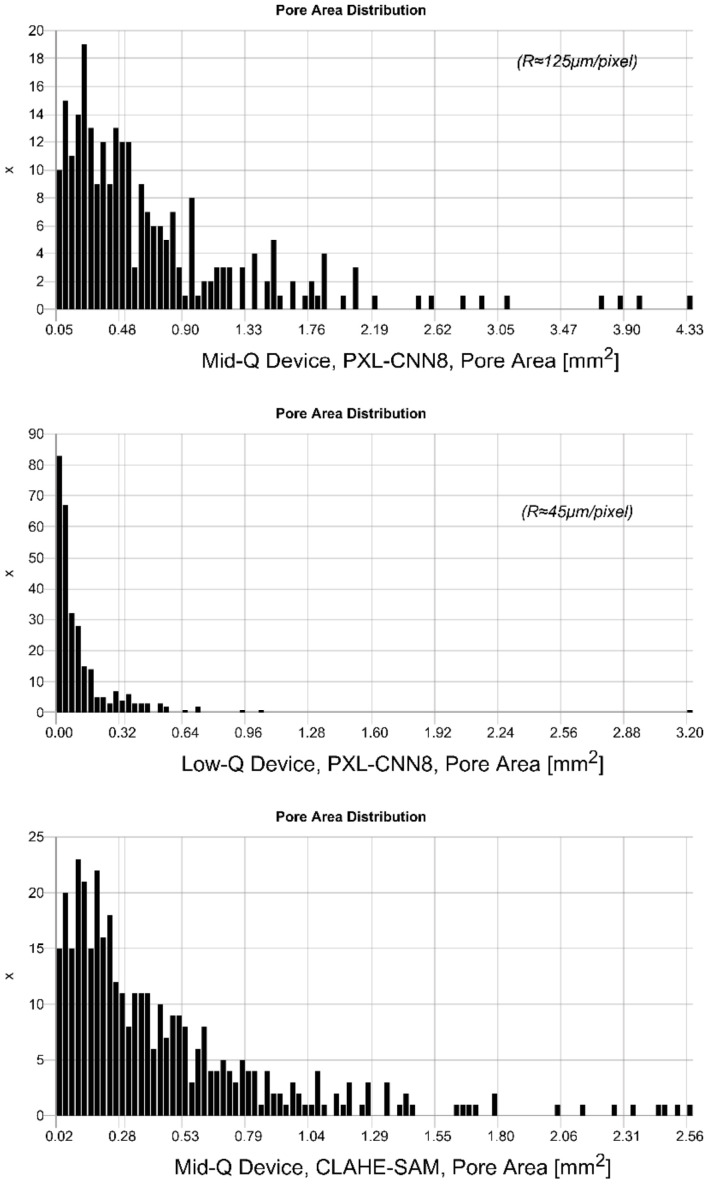
Comparison of pore area distribution analysis from Mid-Q and Low-Q devices (Low-Q poses higher noise, longer exposure times, but higher resolution, too) using the pixel classifier (PXL-CNN8) with one convolution layer and 8 filters, compared with the deep learning CLAHE–SAM model.

**Figure 26 sensors-24-02933-f026:**
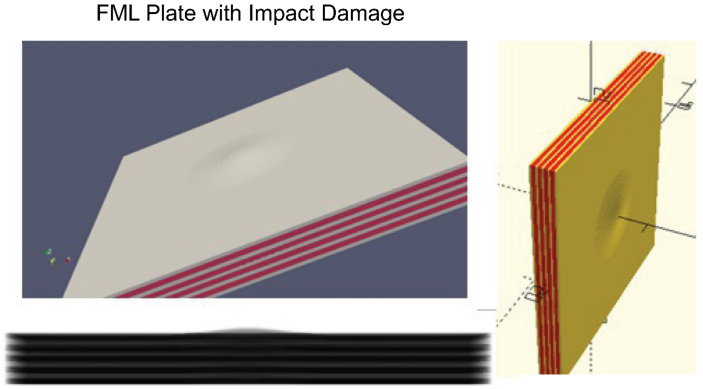
Synthetic CAD model of an FML plate with deformations due to an impact damage.

**Figure 27 sensors-24-02933-f027:**
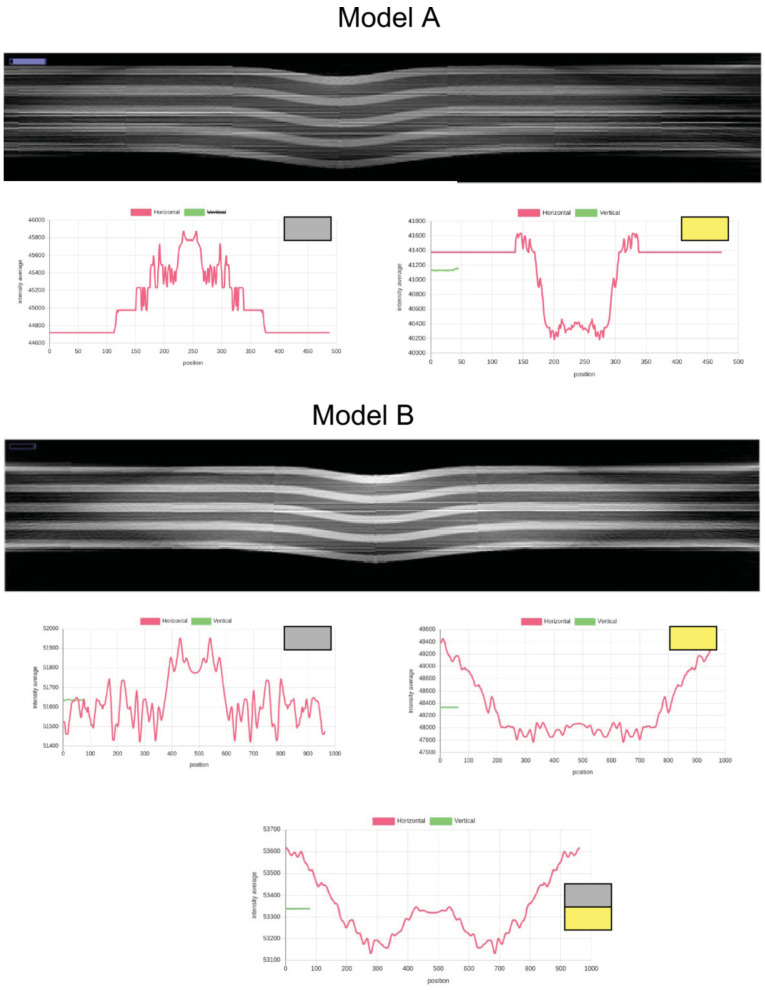
One slice of the CT reconstruction using synthetic X-ray images and intensity profiles for aluminum, PREG, and both layersB. Gray: Aluminum layer, Yellow: PREG layer.

**Table 1 sensors-24-02933-t001:** Material composition of alloys.

Element	Al	Si	Fe	Cu	Mn	Mg	Cr	Ni	Zn	Pb	Sn	Ti
content [wt.%]	bal.	8.6	1	2.1	0.2	0.22	0.04	0.04	0.6	0.03	0.02	0.04

**Table 2 sensors-24-02933-t002:** Summary of HPDC process simulation.

Casting Simulation Output	Description and Interpretation of Simulation Outcome	Conclusion
Tracer Particles	Primary melt flow reflects the width of the in-gates, which falls short of the total width of the sample. Eddies and turbulent flow possible in the lateral parts of the sample.	Increased probability of gas porosity, characterized by high sphericity values, at both sides of the casting. Superimposed slight increase at either end also possible in central cross sections.
Melt Velocity	Absolute velocity at later stages of mold filling reflects findings based on tracer particles, showing reduced velocity at the sides of the sample.	
Entrapped Air	Prediction of mass of air entrapped within the casting per unit volume, based on simulation of flow patterns.	
Porosity	Probability of the occurrence of shrinkage porosity throughout the casting. Probability above zero limited to the overflows. No information about the actual casting.	No information on porosity distribution within the casting.
Hot Spot FS Time	Hot spots predicted within the core of the casting. Decreasing cross-sectional area with increasing distance from in-gates and biscuit.	Increased likelihood of shrinkage porosity, characterized by low sphericity values, within the central area of the plate. Potential decrease in the respective porosity values with increasing distance from the in-gates. No quantitative estimate of porosity levels is available.

**Table 4 sensors-24-02933-t004:** Statistics of the CT pore analysis from the first 500 largest pores (center positions *c* and ellipsoid sizes *s* in mm units, angles *a* in degree, with respect to the *x*-, *y*-, and *z*-axis, area in mm^2^ and volume in mm^3^).

stat\var	x_c_	y_c_	z_c_	x_s_	y_s_	z_s_	x_a_	y_a_	z_a_	Area	Volume
min	0.27	0.09	0.04	0.07	0.09	0.09	0.97	3.00	0.84	0.05	0.01
q1	13.18	0.74	7.92	0.14	0.16	0.20	34.93	63.59	58.95	-	-
median	18.43	0.93	14.43	0.18	0.21	0.27	61.45	89.60	85.93	0.10	0.03
mean	18.73	0.93	14.51	0.22	0.26	0.34	86.14	89.59	84.30	0.12	0.03
q3	24.42	1.10	21.90	0.26	0.30	0.41	142.28	115.71	110.11	-	-
max	36.89	1.77	28.14	1.18	1.62	2.41	178.22	175.81	179.36	2.16	0.71

**Table 5 sensors-24-02933-t005:** Typical layer architecture parameters of a pixel classifier.

Layer	Type	Parameter
1	Input	input = [20,20], output = [20,20]
2.1	Convolution	filters = 8, kernel.size = [5,5], padding = 2, stride = 1, output = [20,20,8]
2.2	Map(Relu)	output = [20,20,8]
2.3	Pooling	kernel.size = [2,2], filters = 8, stride = 2, output = [10,10,8]
3.1opt	Convolution	filters = 4, kernel.size = [5,5], padding = 2, stride = 1, output = [20,20,4]
3.2opt	F(Relu)	output = [20,20,4]
3.3opt	Pooling	kernel.size = [2,2], filters = 8, stride = 2, output = [10,10,4]
4	FC-NN	input = [800/400], output = [2], activation = sigmoid
5	Softmax	input = [2], output = [2]

## Data Availability

Data is available on request from the corresponding author.

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
