# Peer review of "Automated Porosity Characterization for Aluminum Die Casting Materials Using X-ray Radiography, Synthetic X-ray Data Augmentation by Simulation, and Machine Learning"

_sensors, 2024, doi:10.3390/s24092933_

Round 1

Reviewer 1 Report

Comments and Suggestions for Authors

Dear Authors,

Thank you for submitting your paper to Sensors. In the article, you describe the possibility of using X-ray radiography as an alternative to computed tomography. To obtain reasonable results you applied the neural network and deep learning. In general, your article is very interesting but too long. During reading, I found some problems.

In lines 42-44, you use undefined units for forces. Does the "t" stand for tones? The same line 337.

Why did you use references in so unsorted way? You start with position 27. As a result, there are no references to positions 1 and 6.

Line 103 you can remove the word "too"

You use many abbreviations so it would be nice to create the list of abbreviations. Some abbreviations are redefined in the text.

Line 173 there should be "are" instead of "is".

Line 309 something wrong went with the sentence "is ... is".

Line 340: Fig.X does not exist.

Line 386: You just mentioned that you used the Monte Carlo method but there are no details.

Listing of algorithms is not necessary. They do not include any new knowledge.

The article is too long you can shorten it by cutting the description of OpenSCAD and problems with die-casting. In my opinion, the whole section 2 can be shortened very much.

Line 444: you give the time required for mesh-grid transformation but without the parameters of the computer used it tells nothing.

How did you evaluate the values of parameters in eq.1

Line 505: something wrong is in the sentence ("and can not be added it like".

Line 515: the photon conversion factor is  4.6x10^-4 :)

Eqs. 6 are not explained.

Line 668: there should be "improve".

Line 680: there should be "i.e.".

Line 797: what does "am" stand for?

Quality of figs. 13-15, 25 is too low.

Line 805: remove parenthesis.

Line 902: problem with sentence "The pore are distributions".

Line 908: "if" or "of"?

Line 945: symmetric.

Line 949: "half"???

Line 968: problem with sentence "could be accurately be"

Line 979: gigacasting (starts with a small letter and a single period at the end).

The conclusion that "the simulation of die-casting processes neither provides sufficiently detailed quantitative nor qualitative results of statistical and geometrical pore distributions inside the materials" is rather obvious.

Line 1003: showed???

Line 1005: remove "are"

Line 1010: "were" instead "was"

Best regards,

Comments on the Quality of English Language

Due to many problems with language, the article requires proofreading. Some of the problems are listed in the comments.

Author Response

(see attachement, too)

Dear Authors,

Thank you for submitting your paper to Sensors. In the article, you describe the possibility of using X-ray radiography as an alternative to computed tomography. To obtain reasonable results you applied the neural network and deep learning. In general, your article is very interesting but too long. During reading, I found some problems.

> I would object to the verdict that the article is too long, the more so since the reviewer does not hint at specific information that could be omitted. The problem we describe is complex, and for the reader to be able to judge our work, we believe we should explain both the techniques used as well as the characteristics of the samples we need to expect. Thus we believe that, for example some background on the HPDC process should also be given. We hope the reviewer will follow us in this.

In lines 42-44, you use undefined units for forces. Does the "t" stand for tones? The same line 337.

> The locking force of HPDC machines is usually given in tons, so yes, t stands for tons. We have changed this in the text.

Why did you use references in so unsorted way? You start with position 27. As a result, there are no references to positions 1 and 6.

> Reference numbering has been adapted to the order of appearance of the references in the text.

Line 103 you can remove the word "too"

You use many abbreviations so it would be nice to create the list of abbreviations. Some abbreviations are redefined in the text.

> Abbreviation list is added in appendix A

Line 173 there should be "are" instead of "is".

Line 309 something wrong went with the sentence "is ... is".

Line 340: Fig.X does not exist.

Line 386: You just mentioned that you used the Monte Carlo method but there are no details.

> The MC method is now described in more detail in Sec. 3

Listing of algorithms is not necessary. They do not include any new knowledge.

> The algorithms, i.e., the geometrical construction using CSG, are essential for the modeling of the defects and cannot be omitted. There are multiple construction principles that can be used for modeling defects, here one specific was chosen and implemented.

The article is too long you can shorten it by cutting the description of OpenSCAD and problems with die-casting. In my opinion, the whole section 2 can be shortened very much.

> OpenSCAD is an essential component due to the CSG model, which is described, but there is no detailed description of the OpenSCAD software itself; the die casting simulation is essential for some qualitative results for CAD modeling (including ground truth data generation) and for the interpretation of feature marking in real measuring images w/o any ground truth annotation (plausibility checks). On one hand, due to the lack of porosity results from the die casting simulation we justify our experimental data-driven pore analysis approach. On the other hand, we need some qualitative derivations from the die casting simulation, e.g., pore size probabilities and expected geometrical distribution (pore creation probability. We believe the section on problems with die-casting to be of interest for several reasons: First, it gives an indication of the type of defects to be expect and their approximate locations. We need this to check the plausibility of the results we get from X-ray and CT data as well as defect identification. Furthermore, leaving the link to HPDC out will render it next to impossible for the respective community to find our work, and benefit from it.

Line 444: you give the time required for mesh-grid transformation but without the parameters of the computer used it tells nothing.

> Parameters of the hardware were added

How did you evaluate the values of parameters in eq.1

> The parameters are set or given from CAD model (X-ray energy, absorption coefficients...)

Line 505: something wrong is in the sentence ("and can not be added it like".

Line 515: the photon conversion factor is  4.6 :)

Eqs. 6 are not explained.

> Fixed

Line 668: there should be "improve".

Line 680: there should be "i.e.".

Line 797: what does "am" stand for?

Quality of figs. 13-15, 25 is too low.

> We tried to improve the quality, but some graphics are pixel images produced by software without control of image quality

Line 805: remove parenthesis.

Line 902: problem with sentence "The pore are distributions".

Line 908: "if" or "of"?

Line 945: symmetric.

Line 949: "half"???

Line 968: problem with sentence "could be accurately be"

Line 979: gigacasting (starts with a small letter and a single period at the end).

The conclusion that "the simulation of die-casting processes neither provides sufficiently detailed quantitative nor qualitative results of statistical and geometrical pore distributions inside the materials" is rather obvious.

> This is not meant a conclusion, but rather a statement stressing the importance of our work, and of future work activities in the field. We agree that the formulation is not ideal and have changed it. The conclusions is now described more precisely and we distinguish between quantitative and qualitative or predictive  results (based on physical assumptions).  

Line 1003: showed???

Line 1005: remove "are"

Line 1010: "were" instead "was"

> Fixed

Best regards,

Reviewer 2 Report

Comments and Suggestions for Authors

This article successfully detected and characterized hidden defects, impurities, and damage in homogeneous materials such as aluminum die-casting materials, as well as composite materials such as fiber metal laminates (FML), and has great application prospects. However, some modifications are still needed before publication.

1.      The introduction needs to be improved. The content for High Pressure Die Casting (HPDC), such as advantages and applications, needs to be streamlined; there should not be a large number of method descriptions and pictures in the introduction, e.g., lines 103-135, a simple description is sufficient. Please summarise the innovations and methods of this paper in one paragraph of text rather than spreading them out. The introduction of this paper reads more like the introduction of a dissertation, so please streamline the content as much as possible.

2.      The authors introduce High Pressure Die Casting (HPDC) in the introduction, but the relevant content reappears on lines 178-190. It could be edited appropriately to shorten the length of the article. In line 191-257, the introduction of pores is too much, which can be appropriately deleted. Same problem as the introduction section, the second section is overloaded and needs to be streamlined. The content may now be more appropriate for a dissertation.

3.      In section 3.1, the procedure at lines 396-414 can be deleted as it is not essential.

4.      For the model in Fig. 3, the parameters of the model are not given in the text, and in particular the dimensions of the defects are not stated. or to remind the reader where they appear in the text.

5.      The clarity of figures 5, 6, 7 and 22 are too low, please provide clearer images.

6.      Image pre-processing is done in section 6, can you give a comparison picture before and after image processing.

7.      Section 8.1 does not have to appear in the main text. Because the content of the simulations described is not directly related to the research methodology of this paper, and only a short description of the shortcomings of the simulation methodology is needed, this section can be deleted.

8.      In Section 8, the method proposed in this paper successfully identifies pore defects, but there is no statement related to the identification accuracy. The authors could have added a description of the defect identification accuracy to the results.

9.      The outlook section can be briefly explained, preferably without taking up a lot of space.

Author Response

(see attachement, too)

This article successfully detected and characterized hidden defects, impurities, and damage in homogeneous materials such as aluminum die-casting materials, as well as composite materials such as fiber metal laminates (FML), and has great application prospects. However, some modifications are still needed before publication.

1.      The introduction needs to be improved. The content for High Pressure Die Casting (HPDC), such as advantages and applications, needs to be streamlined; there should not be a large number of method descriptions and pictures in the introduction, e.g., lines 103-135, a simple description is sufficient. Please summarise the innovations and methods of this paper in one paragraph of text rather than spreading them out. The introduction of this paper reads more like the introduction of a dissertation, so please streamline the content as much as possible.

> The introduction was completely revised, shortened, some parts were moved to other sections, and the main methods and novelties were clearly outlined. The initial half page on HPDC is necessary to motivate the research conducted here. We are not sure whether by “streamlining” you actually mean “shortening”. We believe that the current size of this section is not critical in terms of length. Nevertheless we have adapted the text and hope you will appreciate our changes. We see an introduction of this kind as very important, as it directly addresses the community we are targeting with our approach.

2.      The authors introduce High Pressure Die Casting (HPDC) in the introduction, but the relevant content reappears on lines 178-190. It could be edited appropriately to shorten the length of the article. In line 191-257, the introduction of pores is too much, which can be appropriately deleted. Same problem as the introduction section, the second section is overloaded and needs to be streamlined. The content may now be more appropriate for a dissertation.

> We believe this section to be necessary because it provides a background for evaluating the pore detection results. However, we understand the position of the reviewer and have removed some of the more general remarks, moving some of them to the introductory section.

3.      In section 3.1, the procedure at lines 396-414 can be deleted as it is not essential.

> The algorithms, i.e., the geometrical construction using CSG, are essential for the modeling of the defects and cannot be omitted since they introduce a simplification and mesh-grid approximation errors. There are multiple construction principles that can be used for modeling defects, here one specific was chosen and implemented.

4.      For the model in Fig. 3, the parameters of the model are not given in the text, and in particular the dimensions of the defects are not stated. or to remind the reader where they appear in the text.

> Details were added

5.      The clarity of figures 5, 6, 7 and 22 are too low, please provide clearer images.

6.      Image pre-processing is done in section 6, can you give a comparison picture before and after image processing.

> A demonstration chart was added

7.      Section 8.1 does not have to appear in the main text. Because the content of the simulations described is not directly related to the research methodology of this paper, and only a short description of the shortcomings of the simulation methodology is needed, this section can be deleted.

> We strongly disagree. The simulation results provide an indication of the location of different types of pores, showing that gas and shrinkage porosity can be expected in different areas of the cast parts under scrutiny. Since gas and shrinkage porosity are distinguished by different geometrical features, this information serves as contribution for a qualitative verification of pore detection algorithms.

8.      In Section 8, the method proposed in this paper successfully identifies pore defects, but there is no statement related to the identification accuracy. The authors could have added a description of the defect identification accuracy to the results.

> identification accuracy can only be given for the ground truth synthetic X-ray images. The values are already given in Sec. 8.2.2 (Assuming a score threshold of 0.9, the false-positive (FP) rate is (for the initially trained model) below 1% of all pixels, and the false-negative (FN) rate is below 1% of all pores.).

9.      The outlook section can be briefly explained, preferably without taking up a lot of space.

> The outlook is here a description of ongoing work, actually not giving final, but preliminary results, so we provide concepts and stressing the issues thar arise with multi-layer composite and laminate materials.

# Changes

- Revision of introduction wrt. to clarification and focusing on relevant work

- Discussion and comparison of X-ray imaging versa GUW measurements are nor discussed in  new Sec. X-ray measuring methods and devices

- Movement of paragraphs containing details about die casting processes from introduction to die casting simulation section

- Replacement of all die casting simulation figures with enhanced versions (Fig. 12-15), clarifications of caption text.

- Shortening and revision of the introduction section, especially wrt to the die-casting process and simulation

- References were added [BAR00], [WRI97], [ZAC77], [SEK21], [XU11], [BRA07]

- Extension and clarification of caption text Fig. 5,6, clarification of the impact of noise on the signal quality based on different statistical distributions.

- Clarification of the necessity of the die casting process simulation for the structural material simulations with defects (with limited qualitative reference).

- Parts of the HPDC simulation description were moved from the introduction to the HPDC and results sections.

- Monte Carlo simulation method was explained in detail (Sec. SDG)

- Added summary table of HPDC process simulation added (Sec. HPDC) and cross-link to CAD-CSG/X-ray simulation

- Added examples of image processing algorithms applied to CT sample image

- Fig. 1 (overview of methodologies) revised and extended

- The link between HPDC process simulation and structural modeling of defects for X-ray image simulation is clarified. 

- Extension and revision of SAM description, training, and experiments including performance metrics 

Round 2

Reviewer 1 Report

Comments and Suggestions for Authors

Dear Authors,

I can see that you sacrificed a lot of time and work to improve your article. As a result, the paper swelled to 43. For me, it is an unacceptable size. Someone, who cannot describe their results within 20 pages is not ready to publish them. You had time to rethink the structure of the article. However, I will accept that and leave the problem of the size to the Editor.

Best regards,